# PROVABLE REPRESENTATION WITH EFFICIENT PLANNING FOR PARTIALLY OBSERVABLE REINFORCEMENT LEARNING

## ABSTRACT

In real-world reinforcement learning, state information is often only partially observable, which breaks the Markov decision process assumption and leads to inferior performance for algorithms that conflate observations with state. Partially Observable Markov Decision Processes (POMDPs), on the other hand, provide a general framework that allows for partial observability to be accounted for in *learning, exploration and planning*, but presents significant computational and statistical challenges. To address these difficulties, we develop a representation-based perspective that leads to a coherent framework and tractable algorithmic approach for practical reinforcement learning from partial observations.

## 1 INTRODUCTION

Reinforcement learning (RL) addresses the problem of making sequential decisions that maximize a cumulative reward through interaction and observation in an environment (Mnih et al., 2013; Levine et al., 2016). The Markov decision process (MDP) has been the standard mathematical model used for most RL algorithm design. However, the success of MDP-based RL algorithms (Uehara et al., 2021; Zhang et al., 2022; Ren et al., 2023c) relies on an assumption that state information is fully observable, which implies that the optimal policy is memoryless, *i.e.*, optimal actions can be selected based only on the current state (Puterman, 2014). However, such an assumption typically does not hold in practice. For example, in (Mnih et al., 2013; Jiang et al., 2021) only images and dialogues are observed, from which the state information only can be partially inferred. The violation of full observability can lead to significant performance degeneration of MDP-based RL algorithms.

The Partially Observed Markov Decision Process (POMDP) (Åström, 1965) has been proposed to extend the the classical MDP formulation by introducing observation variables that only give partial information about the underlying latent state (Hauskrecht & Fraser, 2000; Roy & Gordon, 2002; Chen et al., 2016). This extension greatly expands applicability of POMDPs over MDPs, but the additional uncertainty modeling creates a non-Markovian dependence between successive observations, even though Markovian dependence is preserved between latent states. Consequently, the optimal policy for a POMDP is no longer memoryless but *entire-history* dependent, expanding state complexity exponentially w.r.t. horizon length. Such a non-Markovian dependence creates a significant computational and statistical complexity challenges in *learning, exploration* and *planning*. In fact, without additional assumptions, computing an optimal policy for a given POMDP with known dynamics (*i.e.*, planning) is PSPACE-complete (Papadimitriou & Tsitsiklis, 1987), while the sample complexity of learning for POMDPs grows exponentially w.r.t. the horizon (Jin et al., 2020a).

Despite the worst case hardness of POMDPs, due to their importance there has been extensive work on developing practical RL algorithms that can cope with partial observations. One common heuristic is to extend MDP-based RL algorithms by maintaining a history window over observations to encode a policy or value function, *e.g.*, recurrent neural networks (Wierstra et al., 2007; Hausknecht & Stone, 2015; Zhu et al., 2017). Such algorithms have been applied to many real-world applications with image- or text-based observations (Berner et al., 2019; Jiang et al., 2021), sometimes even surpassing human-level performance (Mnih et al., 2013; Kaufmann et al., 2023).

Such empirical successes have motivated investigation into *structured* POMDPs that allow some of the core computational and statistical complexities to be overcome, which provides an improved understanding of exploitable structure, and practical new algorithms with rigorous justification. For example, the concept of *decodability* has been used to express POMDPs where the latent state can

be exactly recovered from a window of past observations (Efroni et al., 2022; Guo et al., 2023). *Observability* is another special structure, where the $m$-step emission model is assume to be full-rank, allowing the latent state to be identified from $m$ future observation sequences (Jin et al., 2020a; Golowich et al., 2022; Liu et al., 2022; 2023). Such structures eliminate unbounded history dependence, and thus, reduce the computational and statistical complexity. However, most works rely on the existence of an ideal computational oracle for planning, which, unsurprisingly, is infeasible in practice. Although there have been a few attempts to overcome the computational complexity of POMDPS, these algorithms are either only applicable to the tabular setting (Golowich et al., 2022) or rely on integrations that quickly become intractable for large observation spaces (Guo et al., 2018).

*How can we design an **efficient** and **practical** RL algorithm for **structured** partial observations?*

By "efficient" we mean the statistical and computational complexity avoids an exponential dependence on history length, while the computational components of *learning, planning* and *exploration* are computationally feasible; while by "practical" we mean that every component of an algorithm can be easily implemented and applied to a real-world problem. In this paper, we provide a affirmative answer to these questions. More specifically,

- We show that a linear structured POMDP admits a sufficient representation, *Multi-step Latent Variable Representation ($\mu$LV-Rep)*, that supports exact and tractable representation of the value function (Section 4.1), breaking the computational barriers explained in Section 3.
- We design a computationally efficient planning algorithm that can implement both the principles of optimism and pessimism in the face of uncertainty for online and offline POMDPs, respectively, upon the learned sufficient representation $\mu$LV-Rep (Section 4.2).
- We provide a theoretical analysis of the sample complexity of the structured POMDP, justifying the efficiency in balancing exploitation versus exploration in Section 5.
- We conduct a comprehensive comparison to current existing RL algorithms for POMDPs on several benchmarks, demonstrating the superior performance of $\mu$LV-Rep (Section 7).

## 2 PRELIMINARIES

We follow the definition of partially observable Markov decision process (POMDP) in (Efroni et al., 2022; Liu et al., 2022; 2023), which can be formally denoted as a tuple $\mathcal{P} = (\mathcal{S}, \mathcal{A}, \mathcal{O}, r, H, \rho_0, \mathbb{P}, \mathbb{O})$, where $\mathcal{S}$ represents the state space, $\mathcal{A}$ represents the action space, and $\mathcal{O}$ represents the observation space. The positive integer $H$ denotes the horizon length, $\rho_0$ is the initial state distribution, $r : \mathcal{O} \times \mathcal{A} \to [0, 1]$ is the reward function, $\mathbb{P}(\cdot|s, a) : \mathcal{S} \times \mathcal{A} \to \Delta(\mathcal{S})$ is the transition kernel capturing state dynamics, and $\mathbb{O}(\cdot|s) : \mathcal{S} \to \Delta(\mathcal{O})$ is the emission kernel.

Initially, the agent starts at a state $s_0$ drawn from $\rho_0(s)$. At each step $h$, the agent selects an action $a$ from $\mathcal{A}$. This leads to the generation of a new state $s_{h+1}$ following the distribution $\mathbb{P}(\cdot|s_h, a_h)$, and the agent observes $o_{h+1}$ according to $\mathbb{O}(\cdot|s_{h+1})$. The agent also receives a reward $r(o_{h+1}, a_{h+1})$. Since the observations are partially observable, the transition between observations is non-Markovian, which means we need to consider policies $\pi_h : \mathcal{O} \times (\mathcal{A} \times \mathcal{O})^h \to \Delta(\mathcal{A})$ that depend on the entire history, denoted by $\tau_h = \{o_0, a_0, \cdots, o_h\}$. The value associated with policy $\pi = \{\pi_h\}_{h \in [H]}$ with $[H] := \{0, \ldots, H\}$ is defined as $V^\pi = \mathbb{E}_\pi \left[ \sum_{h \in [H]} r(o_h, a_h) \right]$, and the goal is to find the optimal policy $\pi^* = \arg\max_\pi V^\pi$. Note that, the Markov Decision Process (MDP), given by $\mathcal{M} = (\mathcal{S}, \mathcal{A}, r, H, \rho_0, \mathbb{P})$, is a special case of a POMDP, where the state space $\mathcal{S}$ is equivalent to the observation space $\mathcal{O}$, and the emission kernel $\mathbb{O}(o|s)$ is defined as $\delta(o = s)$.

Define the belief $b : \mathcal{O} \times (\mathcal{A} \times \mathcal{O})^h \to \Delta(\mathcal{S})$. Let $b(s_1|o_1) = \mathbb{P}(s_1|o_1)$. We can recursively compute

$$b(s_{h+1}|\tau_{h+1}) \propto \int_{\mathcal{S}} b(s_h|\tau_h)\mathbb{P}(s_{h+1}|s_h, a_h)\mathbb{O}(o_{h+1}|s_{h+1})ds_h.$$

With such definition, one can convert a POMDP to an equivalent belief MDP, denoted as $\mathcal{M}_b = (\mathcal{B}, \mathcal{A}, R_h, H, \mu_b, T_b)$, where $\mathcal{B} \subseteq \Delta(\mathcal{S})$ represents the set of possible beliefs, and

$$\mu_b = \int b(s|o_1)\mu(o_1)do_1, \ \mathbb{P}_b(b_{h+1}|b_h, a_h) = \int \mathbf{1}_{b_{h+1}=b(\tau_h, a_h, o_{h+1})}\mathbb{P}(o_{h+1}|b_h, a_h)do_{h+1}. \quad (1)$$

Here we use $b$ with subscript as one element of $\mathcal{B}$ and use $b$ itself as the mapping $b : \mathcal{O} \times (\mathcal{A} \times \mathcal{O})^h \to \Delta(\mathcal{S})$. We emphasize that each belief corresponds to a density measure defined over the state space. Therefore, $\mathbb{P}_b(\cdot|b_h, a_h)$ represents a conditional operator that characterizes the transition between belief distributions. Considering a policy $\pi : \mathcal{B} \to \Delta(\mathcal{A})$, we can define the state value function

$V_h^\pi(b_h)$ and state-action value function $Q_h^\pi(b_h, a_h)$ for the belief MDP:

$$V_h^\pi(b_h) = \mathbb{E}\left[\sum_{t=h}^H r(o_t, a_t)|b_h\right], \quad Q_h^\pi(b_h, a_h) = \mathbb{E}\left[\sum_{t=h}^H r(o_t, a_t)|b_h, a_h\right]. \quad (2)$$

Adopting the perspective of the equivalent belief MDPs, we can express the Bellman equation as:

$$V_h^\pi(b_h) = \mathbb{E}_\pi\left[Q_h^\pi(b_h, a_h)\right], \quad Q_h^\pi(b_h, a_h) = r(o_h, a_h) + \mathbb{E}_{\mathbb{P}_b}\left[V_{h+1}^\pi(b_{h+1})\right]. \quad (3)$$

Although it is still feasible to utilize a dynamic programming approach and apply the Bellman equation (3) to solve a POMDP, it is crucial to recognize that the dependence of the belief on either the *entire history* results in the possibility of a large or the infinite number of beliefs, even when the number of states is finite, hence leading to infeasible computational and statistical complexity (Papadimitriou & Tsitsiklis, 1987; Jin et al., 2020a). Incorporating function approximation into the learning and planning in a POMDP is significantly more involved than in an MDP.

Consequently, several special structures has been introduced to reduce the statistical complexity of a POMDP. Specifically, *L-decodability* and *$\gamma$-observability* have been introduced in (Du et al., 2019; Efroni et al., 2022) and (Golowich et al., 2022; Even-Dar et al., 2007), respectively.

**Definition 1** (*L*-step decodability (Efroni et al., 2022))**.** $\forall h \in [H]$, *define*

$$x_h \in \mathcal{X} := (\mathcal{O} \times \mathcal{A})^{L-1} \times \mathcal{O}, \quad x_h = (o_{h-L+1}, a_{h-L+1}, \cdots, o_h), \quad (4)$$

*and there exists a decoder* $p^* : \mathcal{X} \to \Delta(\mathcal{S})$, *such that* $p^*(x_h) = b(\tau_h)$.

**Definition 2** ($\gamma$-observability (Golowich et al., 2022; Even-Dar et al., 2007))**.** *Denote* $\langle \mathbb{O}, b \rangle := \int \mathbb{O}_h(\cdot|s) b(s) ds$, *for arbitrary beliefs* $b$ *and* $b'$ *over states,* $\|\langle \mathbb{O}, b \rangle - \langle \mathbb{O}, b' \rangle\|_1 \geqslant \gamma \|b - b'\|_1$.

Note that we slightly generalize decodability from Efroni et al. (2022), which assumes $b(\tau_h)$ is a Dirac measure on $\mathcal{S}$. It is worth noticing that $\gamma$-observability and *L*-decodability are highly related. Existing works have shown that a $\gamma$-observable POMDP can be well approximated by a decodable POMDP with a history of proper length $L$ (Golowich et al., 2022; Uehara et al., 2022; Guo et al., 2023) (see Appendix B for a detailed discussion). Hence, in the main text, we focus on algorithm design for an *L*-decodable POMDP, which can be directly extended to a $\gamma$-observable POMDP.

## 3 DIFFICULTIES IN POMDP PLANNING FROM A REPRESENTATION VIEW

Obviously, such structures for a POMDP reduce the history dependence, and thus reduce statistical complexity. However, the computational tractability of planning and exploration given such structures remains open. Before attempting a practical algorithm for a structured POMDP, we first explain the the benefits of a linear structure representation, and the challenges in being applied for POMDPs.

**Linear Structure of MDPs.** Linear structures for MDPs were introduced in (Jin et al., 2020b; Yang & Wang, 2020; Ren et al., 2022; Uehara et al., 2021) to enable effective function approximation and address the core computational challenges of planning and exploration in general nonlinear control. Such an approach leverages the spectral factorization of transition kernel and reward, given by:

$$\mathbb{P}(s'|s, a) = \langle \phi(s, a), \mu(s') \rangle_{\mathcal{H}}, \ r(s, a) = \langle \phi(s, a), \theta \rangle_{\mathcal{H}}, \quad (5)$$

where $\phi : \mathcal{S} \times \mathcal{A} \to \mathcal{H}$ and $\mu : \mathcal{S} \to \mathcal{H}$ are feature maps to a Hilbert space $\mathcal{H}$. Under this factorization, the state-action value function $Q^\pi$ for an arbitrary policy $\pi$ can be represented as:

$$Q_h^\pi(s, a) = r(s, a) + \int V_{h+1}^\pi(s')\mathbb{P}(s'|s, a)ds' = \left\langle \phi(s, a), \theta + \int_{\mathcal{S}} V_{h+1}^\pi(s')\mu(s')ds' \right\rangle_{\mathcal{H}}, \quad (6)$$

where the first equation comes from the Bellman recursion, and the second equation is obtained by plugging in (5). This result implies that instead of dealing with a complex function space defined on the raw state space, one can design computationally efficient planning and exploration algorithms in the space linearly spanned by $\phi$. In fact, based on the correspondence between policies and $Q$-functions discussed in Ren et al. (2023c), $\phi$ can be interpreted as representing primitives for constructing a skill set. Zhang et al. (2022); Qiu et al. (2022) took initial strides in developing efficient and practical algorithms that harness the linear structure of MDPs. Building upon their work, Ren et al. (2023b) observed that the transition kernel can be formulated with a latent variable model:

$$\mathbb{P}(s'|s, a) = \int_{\mathcal{Z}} p(z|s, a)p(s'|z)d\mu = \langle p(\cdot|s, a), p(s'|\cdot) \rangle_{L_2(\mu)} \quad (7)$$

with the linear structure over conditional distributions $p(\cdot|s, a) \in L_2(\mu)$, $p(s'|\cdot) \in L_2(\mu)$ for the Lebesgue measure $\mu$. This linear structure can be leveraged to design a practical representation learning algorithm similar to the Dreamer-style algorithm (e.g. Hafner et al., 2021), but with more theoretically sound planning and exploration mechanisms.

**Difficulties for POMDPs.** Inspired by the successes of linear structure for MDPs, it is natural to consider an extension to POMDPs. However, limited progress had been made in exploiting the linear

structure of POMDPs (Guo et al., 2023). Specifically, for an arbitrary policy $\pi$, consider the Bellman equation:

$$Q_h^\pi(b_h, a_h) = r(o_h, a_h) + \mathbb{E}_{\mathbb{P}(o_{h+1}|b_h, a_h)}\left[V_{h+1}^\pi(b(\tau_h, a_h, o_{h+1}))\right] \tag{8}$$

The second equation comes from the belief transition (1). Straightforwardly applying the linear structure in the latent state transition, we obtain

$$Q_h^\pi(b_h, a_h) = r(o_h, a_h) + \mathbb{E}_{b_h(s)}\left[\int \mathbb{P}(s_{h+1}|s_h, a_h)\mathbb{E}_{O(o_{h+1}|s_{h+1})}\left[V_{h+1}^\pi(b(\tau_h, a_h, o_{h+1}))\right]\right] \tag{9}$$

$$= r(o_h, a_h) + \left\langle \int {\color{red}b_h(s)}\phi(s, a)\,ds, \int \mu(s_{h+1})\,\mathbb{E}_{O(o_{h+1}|s_{h+1})}\left[V_{h+1}^\pi({\color{red}b(\tau_h, a_h, o_{h+1})})\right]ds_{h+1}\right\rangle.$$

From (9), one can see the major computational difficulties marked red as:

i) the representation is $\int {\color{red}b_h(s)}\phi(s, a)\,ds$, which requires not only belief, but also an *integration* with an unknown $\phi(s, a)$ upon the belief, therefore, is difficult to estimate; more importantly,

ii) the ${\color{red}b(\tau_h, a_h, o_{h+1})}$ inside the *nonlinear* $V_{h+1}^\pi(\cdot)$ depends on the history, hence the integration will be a function of history, rather than a vector, which breaks the linear structure of $Q_h^\pi$.

These difficulties have made the usage of linear structure in POMDPs highly non-trivial, even with $L$-step decodability (Guo et al., 2023).

## 4 MULTI-STEP LATENT VARIABLE REPRESENTATION

We now discuss how to leverage the structure of $L$-step decodable POMDPs to bypass the two difficulties revealed above, and eventually develop a **computationally efficient** planning algorithm. Our key observation is that, the value function for an $L$-step decodable POMDP only depends on the last $L$-step observations, and for a policy family, they can be *linearly* represented by an $L$-step *latent variable representation*, therefore, bypass the beliefs. We then show how to design a latent variable representation learning and planning algorithm upon (Ren et al., 2023b), which leads to *Multi-step Latent Variable Representation ($\mu$LV-Rep)*.

### 4.1 KEY OBSERVATIONS

Although the equivalent belief MDP provides a Markovian Bellman recursion (8), as we discussed in Section 3, the linear structure does not directly introduce tractability in planning.

To resolve such difficulties in a belief MDP, we make a first key observation, *i.e.*, **an observation-based value function will bypass the necessity for belief computation.** Specifically, $b_h(\cdot)$ is a mapping from the history $\tau_h$ to the space of probability densities over state. Therefore, we rewrite $Q_h^\pi(b_h, a_h) = Q_h^\pi(\tau_h, a_h)$. With this simply reformulation, we avoid explicit dependence on beliefs in the $Q^\pi$-function, eliminating the first difficulty.

Next, we make a second key observation, *i.e.*, **by the definition of $L$-step decodability in a POMDP, it is sufficient to consider the $L$-step memory $x_h$, instead of the entire history $\tau_h$.** This can be easily verified from the $L$-step decodability definition 1. Specifically, because of $L$-step decodability, the beliefs $b_h(s)$ can be represented with a decoder from $L$-step windows over $x_h$, although the decoder is unknown, leading to $Q_h^\pi(\tau_h, a_h) = Q_h^\pi(x_h, a_h)$. Directly inserting $Q_h^\pi(x_h, a_h)$ into the Bellman equation (8) yields

$$Q_h^\pi(x_h, a_h) = r(o_h, a_h) + \mathbb{E}_{\mathbb{P}^\pi(x_{h+1}|x_h)}\left[V_{h+1}^\pi(x_{h+1})\right]. \tag{10}$$

This outcome reduces the statistical complexity, as previously exploited in (Efroni et al., 2022; Guo et al., 2023). However, the second difficulty remains, *i.e.*, there is an additional dependence of $V_{h+1}^\pi(x_{h+1})$ on $(x_h, a_h)$, since $x_{h+1} = (o_{h-L+2}, a_{h-L+2}, \cdots, o_{h+1})$ has an overlap with $(x_h, a_h) = (o_{h-L+1}, a_{h-L+1}, o_{h-L+2}, a_{h-L+2}, \cdots, o_h, a_h)$. This overlap between the successive $x_h$ and $x_{h+1}$, which are also known as mega-states (Efroni et al., 2022), breaks the low-rank structure, and thus, impedes directly extending low-rank MDP for POMDPs.

Recall that by $L$-step decodability, $V_{h+L}^\pi(x_{h+L})$ will be independent of $(x_h, a_h)$. Therefore, our first attempt is to consider the $L$-step Bellman equation for $Q_h^\pi(\tau_h, a_h)$, which can be easily derived by expanding (8)

$$Q_h^\pi(x_h, a_h) = \mathbb{E}_{\pi_{h+1:h+L}|x_h, a_h}\left[\sum_{i=h}^{h+L-1} r(o_i, a_i) + V_{h+L}^\pi(x_{h+L})\right]. \tag{11}$$

At first glance, the $L$-step forward expansion induces $V_{h+L}^\pi(x_{h+L})$, which eliminates the overlapping dependence of $x_{h+L}$ on $(x_h, a_h)$, due to the $L$-step decodability. However, the remaining issue is that the policy $\pi_{h+1:h+L-1}$ still depends on some part of $(x_h, a_h)$, which retains a dependence on $\mathbb{E}_{\pi_{h+1:h+L}|x_h, a_h}\left[V_{h+L}^\pi(x_{h+L})\right]$, and thus still breaks the linear structure for the $Q_h^\pi$-function. To

recover a linear structure for the $Q^\pi$-function, we introduce our most important observation, *i.e.*, **if we consider a policy $\nu_\pi$, that conditioned on the sufficient $h$-step latent variable induced by the observation dynamics, as well as the future action-observation sequences, the dependence of $\pi_{h+1:h+L-1}$ on $(x_h, a_h)$ at $L$-step can be eliminated.** Specifically, consider

$$\mathbb{P}^\pi (x_{h+L}|x_h, a_h) = \int p(z_{h+1}|x_h, a_h) \mathbb{P}^{\nu_\pi} (x_{h+L}|z_{h+1}) \, dz_{h+1} = \langle p(\cdot|x_h, a_h), \mathbb{P}^{\nu_\pi} (x_{h+L}|\cdot)\rangle_{L_2(\mu)}, \tag{12}$$

where $z$ denotes the latent variable. This observation to eliminate the policy dependence on $(x_h, a_h)$ has been discussed in (Efroni et al., 2022) as the *"moment matching policy"*. The existence of such sufficient $h$-step latent variable is guaranteed by the low rank structure, while the existence of the equivalent moment matching policy $\nu^\pi$ is guaranteed from $L$-decodability. Note that, the idea of the moment matching policy was only considered as a proof trick, and not previously exploited to reveal linear structure for algorithm design. The concrete moment matching policy is discussed in details in Appendix C.

One key difference between the factorization in (12) and the linear structure in (5) or (7) for an MDP is that in a linear MDP, one obtains a *policy-independent* decomposition, where both the components $\phi(s, a)$ and $\mu(s')$ from the transition dynamics are invariant w.r.t. the policy. Clearly, in (12) for a POMDP, one component from the obtained factorization, $\mathbb{P}^{\nu_\pi}(x_{h+L}|\cdot)$, depends on the policy. However, we will see that this does not affect the linear representation ability of $p(\cdot|x_h, a_h)$ for $Q^\pi$.

**Remark (Identifiability):** It should be noted that we deliberately use $z$ as the latent variable, rather than $s$, in (12) to emphasize the learned latent variable structure can be different from the groundtruth state, hence without an *idenfitiability* assumption. Nevertheless, the learned structure has the same effect in representing $Q^\pi$ linearly.

Now, we have established every component needed to derive the linear representation by introducing (12) into (11). For the first term in (11), for $\forall k \in \{1, \ldots, L-1\}$, we have

$$\mathbb{E}_\pi [r(o_{h+k}, a_{h+k})] = \left\langle p(\cdot|x_h, a_h), \underbrace{\int \mathbb{P}^{\nu_\pi} (o_{h+k}, a_{h+k}|\cdot) r(o_{h+k}, a_{h+k}) \, do_{h+k} da_{h+k}}_{w_k^\pi(\cdot)} \right\rangle. \tag{13}$$

With the "moment matching policy" trick, the $\nu_\pi$ is independent w.r.t. $(x_h, a_h)$. Then, $\mathbb{P}^{\nu_\pi}(o_{h+k}, a_{h+k}|\cdot)$ is independent to history, which leads to the linear representation in (13).

For the second term in (11), similarly, we have

$$\mathbb{E}_\pi \left[ V_{h+L}^\pi (x_{h+L}) \right] = \int \mathbb{P}^\pi (x_{h+L}|x_h, a_h) V^\pi (x_{h+L}) \, dx_{h+L}$$

$$= \left\langle p(\cdot|x_h, a_h), \underbrace{\int \mathbb{P}^{\nu_\pi} (x_{h+L}|\cdot) V^\pi (x_{h+L}) \, dx_{h+L}}_{w_{h+L}^\pi(\cdot)} \right\rangle. \tag{14}$$

Recall that $x_{h+L}$ does not have overlap with $x_h$, with the same "moment match policy" trick, $w_{h+L}^\pi(\cdot)$ is indepenent w.r.t. $(x_h, a_h)$.

Together with (13) and (14), define $w^\pi = \sum_{k=h}^{h+L} w_{h+k}^\pi$, we can justify that under our key observations, for an $L$-step decodable POMDP, $Q^\pi$ can be represented linearly in $p(\cdot|x_h, a_h)$ as

$$Q_h^\pi (x_h, a_h) = \langle p(\cdot|x_h, a_h), w^\pi(\cdot)\rangle_{L_2(\mu)}. \tag{15}$$

With assumption that $r(o_h, a_h) = \langle p(\cdot|x_h, a_h), \omega^r(\cdot)\rangle$, which is easy to achieve by feature augmentation (Ren et al., 2023a).

**Connection to PSR (Littman & Sutton, 2001).** Both the proposed $\mu$LV-Rep and the predictive state representation (PSR) (Littman & Sutton, 2001) bypass the explicit belief calculation by factorizing the observation transition system. However, there are significant differences between the structures, and hence in planning and exploration. Specifically, the PSR is based on the assumption that, for any finite sequence of events $y_{h+1:k} = (o_{h+1:h+k}, a_{h:h+k-1})$ upon history $x_h$ with $k \in \mathbb{N}_+$, the probability can be linearly factorized as $\mathbb{P}(o_{h+1:h+k}|x_h, a_{h:h+k-1}) = \langle \omega_{y_{h+1:k}}, \mathbb{P}(U|x_h)\rangle$, where $\omega_{y_{h+1:k}} \in \mathbb{R}^d$, $U := [u_i]_{i=1}^d$ is a set of core test events, and $\mathbb{P}(U|x_h)$ is referred as the predictive state representation at $h$-step. Then, the forward observation dynamics can be represented in PSR via

---

**Algorithm 1** Online Exploration for $L$-step decodable POMDPs with Latent Variable Representation

1: **Input:** Model Class $\mathcal{M} = \{\{(p_h(z|x_h, a_h), p_h(o_{h+1}|z)\}_{h \in [H]}\}$, Variational Distribution Class $\mathcal{Q} = \{\{q_h(z|x_h, a_h, o_{h+1})\}_{h \in [H]}\}$, Episode Number $K$.

2: **Initialize** $\pi_0^h(s) = \mathcal{U}(\mathcal{A}), \forall h \in [H]$ where $\mathcal{U}(\mathcal{A})$ denotes the uniform distribution on $\mathcal{A}$; $\mathcal{D}_{0,h} = \emptyset, \mathcal{D}'_{0,h} = \emptyset, \forall h \in [H]$.

3: **for** episode $k = 1, \cdots, K$ **do**

4:     Initialize $\mathcal{D}_{k,h} = \mathcal{D}_{k-1,h}, \mathcal{D}'_{k,h} = \mathcal{D}'_{k-1,h}$

5:     **for** Step $h = 1, \cdots, H$ **do**

6:         Collect the transition $(x_h, a_h, o_{h+1}, a_{h+1}, \cdots, o_{h+L-1}, a_{h+L-1}, o_{h+L})$ where $x_h \sim d_{\mathcal{P}}^{\pi_k, h}, a_{h:h+L-1} \sim \mathcal{U}(\mathcal{A}), o_{h+i} \sim \mathbb{P}^{\mathcal{P}}(\cdot|x_{h+i-1}, a_{h+i-1}), \forall i \in [L]$.

7:         $\mathcal{D}_{k,h} = \mathcal{D}_{k,h} \cup \{x_h, a_h, o_{h+1}\}, \mathcal{D}'_{k,h+i} = \mathcal{D}'_{k,h+i} \cup \{x_{h+i}, a_{h+i}, o_{h+i+1}\}, \forall i \in [L]$.

8:     **end for**

9:     Learn the latent variable model $\hat{p}_k(z|x_h, a_h)$ with $\mathcal{D}_{k,h} \cup \mathcal{D}'_{k,h}$ via maximizing the ELBO, and obtain the learned model $\widehat{\mathcal{P}}_k = \{(\hat{p}_{h,k}(z|x_h, a_h), \hat{p}_{h,k}(o_{h+1}|z))\}_{h \in [H]}$.

10:     (Optional) Set the exploration bonus $\hat{b}_{k,h}(s, a)$ with $\mathcal{D}_{k,h}$.

11:     Update policy $\pi_k = \arg\max_\pi V_{\widehat{\mathcal{P}}_k, r + \hat{b}_k}^\pi$.

12: **end for**

13: **Return** $\pi^1, \cdots, \pi^K$.

---

Bayes' rule, *i.e.*, $\mathbb{P}(o_{h+2:k}|x_h, a_{h:h+k-1}, o_{h+1}) = \frac{\langle \omega_{y_{h+2:k}}, \mathbb{P}(U|x_h) \rangle}{\langle \omega_{y_{h+1}}, \mathbb{P}(U|x_h) \rangle}$, which introduces a nonlinear operation, making the planning and exploration difficult.

### 4.2 MAIN ALGORITHM

We have revealed the linear representation for $Q_h^\pi$. In this section, we will discuss how we can learn the representation, and the planning and exploration procedure upon the representation. The full algorithm is presented in Algorithm 1.

**Variational Learning of $\mu$LV-Rep.** As we generally do not have the latent variable representation $p(\cdot|x_h, a_h)$ a priori, it is essential to perform the representation learning with online collected data. One straightforward idea is to apply maximum likelihood estimation on $\mathbb{P}^\pi(x_{h+k}|x_h, a_h)$. Although this is theoretically correct, due to the overlap on $x_{h+k}$ and $x_h$, there will be parametrization issue, with the waste of memory and computation cost. Recall the fact that we only need $p(z_h|x_h)$ for representing $Q_h^\pi$, and the observation that for $\forall k \in \mathbb{N}_+$,

$$p(o_{h+1:h+l}|x_h, a_h) = \int_{\mathcal{Z}} p(z_h|x_h, a_h) \underbrace{\prod_{i=1}^{l} \left[ \int_{\mathcal{Z}} \mathbb{P}^\pi(z_{h+i}|z_{h+i-1}, a_i) p(o_{h+i}|z_{h+i}) dz_{h+i} \right]}_{\mathbb{P}^\pi(o_{h+1:h+l}|z_h)} dz_h, \quad (16)$$

we can obtain $p(\cdot|x_h, a_h)$ by performing maximum likelihood estimation (MLE) on $p(o_{h+1:h+l}|x_h, a_h)$ for arbitrary $l \in \mathbb{N}_+$. We exploit the evidence lower bound (ELBO) (Ren et al., 2023b) for a tractable surrogate of MLE of the latent variable model (16), *i.e.*,

$$\log p(o_{h+1:h+l}|x_h, a_h) = \log \int_{\mathcal{Z}} p(z_h|x_h, a_h) \mathbb{P}^\pi(o_{h+1:h+l}|z_h)$$

$$= \log \int_{\mathcal{Z}} \frac{p(z_h|x_h, a_h) \mathbb{P}^\pi(o_{h+1:h+l}|z_h)}{q(z|x_h, a_h, o_{h+1:h+l})} q(z|x_h, a_h, o_{h+1:h+l}) \quad (17)$$

$$= \max_{q \in \Delta(\mathcal{Z})} \mathbb{E}_{q(\cdot|x_h, a_h, o_{h+1:h+l})} [\log \mathbb{P}^\pi(o_{h+1:h+l}|z_h)] - D_{KL}(q(z|x_h, a_h, o_{h+1:h+l})||p(z_h|x_h)),$$

where the last equation comes from Jensen's inequality, with the equality holds when $q(z|x_h, a_h, o_{h+1:h+l}) \propto p(z_h|x_h, a_h) \mathbb{P}^\pi(o_{h+1:h+l}|z_h)$. One can use (17) with data to fit the $\mu$LV-Rep. For the ease of the presentation, we choose $l = 1$ in Algorithm 1.

**Practical Parametrization of $Q^\pi$ with $\mu$LV-Rep.** With $\mu$LV-Rep, we can represent $Q_h^\pi(x_h, a_h) = \langle p(z|x_h), w_h^\pi(z) \rangle_{L_2(\mu)}$. If the latent variable $z$ in $p(z|x_h)$ is an enumeratable discrete variable, $Q^\pi(x_h, a_h) = \sum_{i=m} w^\pi(z_i) p(z_i|x_h)$, can be simply represented.

However, the discrete latent variable is not differentiable, which may lead to some difficulty in learning. Therefore, continuous latent variable $z$ will be used, which induces infinite-dimensional $w(z)$. We follow the trick in LV-Rep (Ren et al., 2023b) that we $Q^\pi(x_h, z_h)$ as an expectation,

$$Q^\pi(x_h, a_h) = \langle p(z|x_h), w^\pi(z) \rangle = \mathbb{E}_{p(z|x_h)}[w^\pi(z)]$$

which can be either approximated by Monte-Carlo method or random feature quadrature (Ren et al., 2023b), respectively,

$$Q^\pi(x_h, a_h) \approx \frac{1}{m}\sum_{i=1}^m w^\pi(z_i) \quad \text{or} \quad Q^\pi(x_h, a_h) \approx \frac{1}{m}\sum_{i=1}^m \tilde{w}^\pi(\xi_i)\varphi(z_i, \xi_i) \quad (18)$$

with samples $z_i \sim p(z|x_h)$ and $\xi_i \sim P(\xi)$ as the random feature measure for the RKHS space of $w(z)$. Both of these two approximation can be implemented by a neural network. Due to space limitation, we omit the derivation of the random feature quadrature. Please refer to Appendix E.

**Planning and Exploration with $\mu$LV-Rep.** With an accurate estimation of $Q$ function, we can perform planning with the standard dynamic programming approach (e.g. Munos & Szepesvári, 2008). However, dynamic programming involves an $\arg\max$ operations, which can only be possible when $|\mathcal{A}| < \infty$. To deal with the continuous action scenarios, we can leverage the popular policy gradient methods like SAC (Haarnoja et al., 2018), with the critic parameterized with $\mu$LV-Rep.

To improve the exploration, we can leverage the idea of Uehara et al. (2021); Ren et al. (2023b) and add an additional ellipsoid bonus to implement the optimism in the face of uncertainty principle. Specifically, if we use the random feature quadrature, we can compute such bonus via:

$$\hat{\psi}_{h,k}(x_h, a_h) = [\varphi(z_i; \xi_i)]_{i\in[m]}, \quad \text{where} \quad \{z_i\}_{i\in[m]} \sim \hat{p}_{k,h}(z|x_h, a_h), \quad \{\xi_i\}_{i\in[m]} \sim P(\xi),$$

$$\hat{b}_{k,h}(s, a) = \alpha_k \hat{\psi}_{h,k}(x_h, a_h)\hat{\Sigma}_{k,h}^{-1}\hat{\psi}_{h,k}(x_h, a_h),$$

with $\hat{\Sigma}_{k,h} = \sum_{(x_{h,i}, a_{h,i})\in\mathcal{D}_{k,h}} \hat{\psi}_{h,k}(x_{h,i}, a_{h,i})\hat{\psi}_{h,k}(x_{h,i}, a_{h,i})^\top + \lambda I$, and $\alpha_k$, $\lambda$ are user-specified constants. Similarly, the bonus can be used for implementing the pessimism in the face of uncertainty principle in the offline setting, as we discussed in Appendix D, due to space limitation.

## 5 THEORETICAL ANALYSIS

In this section, we provide a formal sample complexity analysis of the proposed algorithm. We start from the following assumptions, that are commonly used in the literature (e.g. Agarwal et al., 2020; Uehara et al., 2021; Ren et al., 2023b).

**Assumption 1** (Finite Candidate Class with Realizability). $|\mathcal{M}| < \infty$ and $\{(p_h^*(z|x_h, a_h), p_h^*(o_{h+1}|z))\}_{h\in[H]} \in \mathcal{M}$. Meanwhile, for all $(p_h(z|x_h, a_h), p(o_{h+1}|z)) \in \mathcal{M}$, $p_h(z|x_h, a_h, o_{h+1}) \in \mathcal{Q}$.

**Assumption 2** (Normalization Conditions). $\forall \mathcal{P} \in \mathcal{M}, (x_h, a_h) \in \mathcal{X} \times \mathcal{A}, \|p_h(\cdot|x_h, a_h)\|_{\mathcal{H}_K} \leqslant 1$ for some kernel $K$. Furthermore, $\forall g : \mathcal{X} \to \mathbb{R}$ such that $\|g\|_\infty \leqslant 1$, we have $\|\int_{\mathcal{X}} p(x_{h+L}|\cdot)g(x_{h+L})dx_{h+L}\|_{\mathcal{H}_K} \leqslant C$.

Now we are able to provide the sample complexity of $\mu$LV-Rep.

**Theorem 3** (PAC Guarantee, Informal version of Theorem 13). *Assume the kernel $K$ satisfies the regularity conditions in Appendix F.1. If we properly choose the exploration bonus $\hat{b}_k(x, a)$, we can obtain an $\varepsilon$-optimal policy with probability at least $1 - \delta$ after we interact with the environments for $N = \text{poly}(C, H, |\mathcal{A}|^L, L, \varepsilon, \log(|\mathcal{M}|/\delta))$ episodes.*

## 6 RELATED WORK

Representation has been exploited in partially observable reinforcement learning, but for different purposes. Vision-based representations (Yarats et al., 2020; Seo et al., 2023) have been designed to extract compact feature from raw pixel observations. We emphasize that this type of observation feature does not explicitly capture dynamics properties, and essentially orthogonal to but naturally compatible with the proposed representation. Many dynamic-aware representation methods have been developed, such as bi-simulation (Ferns et al., 2004; Gelada et al., 2019; Zhang et al., 2020), successor features (Dayan, 1993; Barreto et al., 2017; Kulkarni et al., 2016), spectral representation (Mahadevan & Maggioni, 2007; Wu et al., 2018; Duan et al., 2019), and contrastive representation (Oord et al., 2018; Nachum & Yang, 2021; Yang et al., 2021). The proposed representation for POMDPs is inspired by recent progress (Jin et al., 2020b; Yang & Wang, 2020; Agarwal et al., 2020; Uehara et al., 2022) in revealing low-rank structure in the transition kernel of MDPs, and inducing effective linear representations for the state-action value function for an arbitrary policy. This prior discovery has led to a series of practical and provable RL algorithms in the MDP setting, achieving a delicate balance between learning, planning and exploration (Ren et al., 2022; Zhang et al., 2022; Ren et al., 2023c;b). Although these algorithms demonstrate theoretical and empirical benefits, they rely on the Markovian assumption, hence are not applicable to the POMDP setting we consider here.

There have been several attempts to exploit the low-rank representation in POMDPs to reduce the statistical complexity (Efroni et al., 2022). Azizzadenesheli et al. (2016); Guo et al. (2016) exploits

spectral learning for model estimation without exploration; Jin et al. (2020a) explores within an spectral estimation set; Uehara et al. (2022) builds upon the Bellman error ball; Zhan et al. (2022); Liu et al. (2022) consider the MLE confidence set for low-rank structured models; and (Huang et al., 2023) construct a UCB-type algorithm upon the MLE ball of PSR. However, these algorithms rely on intractable oracles for planning, and there are fewer works that consider exploiting low-rank structure to achieve computationally tractable planning. One exception is (Zhang et al., 2023; Guo et al., 2023), which still includes intractable operations, *i.e.*, infinite-dimensional operations or integrals.

## 7 EXPERIMENTAL EVALUATION

We evaluate the proposed approach on RL tasks with partial observations, constructed based on the OpenAI gym MuJoCo (Todorov et al., 2012) and DeepMind Control Suites (Tassa et al., 2018), as well as Meta-world (Yu et al., 2019). For our implementation, we employ a continuous latent variable model similar to (Hafner et al., 2020), approximating distributions with Gaussians parameterized by their mean and variance. We use $L = 3$, so the representation is learned by making $L$-step predictions. We apply Soft Actor-Critic (SAC) as the planner (Haarnoja et al., 2018), where a multi-step critic objective is also adopted to improve learning efficiency (Hafner et al., 2020; Feinberg et al., 2018). We first compared the proposed algorithm in partially observable setting and image-based setting. We also provided the ablation study in Appendix H.1.

### 7.1 PARTIALLY OBSERVABLE CONTINUOUS CONTROL

The standard MuJoCo tasks from the OpenAI gym and DeepMind Control Suites are not partially observable. To generate partially observable problems based on these tasks, we adopt a widely employed technique of masking velocities within the observations (Ni et al., 2021; Weigand et al., 2021; Gangwani et al., 2020). In this way, it becomes impossible to extract complete decision-making information from a single environment observation, yet the ability to reconstruct the missing observation remains achievable by aggregating past observations. We verify the hardness of this setting by using the LV-Rep algorithm (Ren et al., 2023b), which takes the raw observation (with masked velocities) as input to the networks. The algorithm fails to learn on all tasks, confirming the difficulty caused by partial observability. We also provide the best performance when using the original fully observable states (without velocity masking) as input, denoted by *Best-FO* (Best result with Full Observations). This gives a reference for the best an algorithm can achieve in our tests.

We consider four baselines in the experiments, including two model-based methods Dreamer (Hafner et al., 2020; 2021) and Stochastic Latent Actor-Critic (SLAC) (Lee et al., 2020), and a model-free baseline, SAC-MLP, that concatenates history sequences (past four observations) as input to an MLP layer for both the critic and policy. This simple baseline can be viewed as an analogue to how DQN processes observations in Atari games (Mnih et al., 2013) as a sanity check. We also compare to the neural PSR (Guo et al., 2018). We compare all algorithms after running 200K environment steps. This setup exactly follows the benchmark (Wang et al., 2019), which has been widely adopted in (Zhang et al., 2022; Ren et al., 2023c;b) for fairness. All results are averaged across four random seeds. Table 1 presents all the experimental results, averaged over four random seeds. The results clearly demonstrate that the proposed method consistently delivers either competitive or superior outcomes across all domains compared to both the model-based and model-free baselines. We note that in most domains, $\mu$LV-Rep nearly matches the performance of Best-FO, further confirming that the proposed method is able to extract useful representations for decision-making in partially observable environments.

### 7.2 IMAGE-BASED CONTINUOUS CONTROL

We then evaluate the proposed method on the DeepMind Control Suites and Meta-world to demonstrate capability in complex visual control tasks. We observe that directly learning a robust control representation by predicting future visual observations can be challenging, since images contain redundant information for effective decision-making. Consequently, a more advantageous approach is to first acquire an image representation then learn a latent representation based on this initial image representation. In particular, we employ visual observations with dimensions of $64 \times 64 \times 3$ and apply a masked autoencoder (MAE) with mask ratio 0.75 to learn a representation of these visual observations (He et al., 2022). The MAE is first pre-trained with random trajectories then fine-tuned by the online learning procedure. This produces compact vector representations for the images, which are then forwarded as input to the representation learning method. More implementation details, including network architectures and parameters, are provided in Appendix H.

Table 1: Performance on various continuous control problems with partial observation. All results are averaged across 4 random seeds and a window size of 10K. $\mu$LV-Rep achieves the best performance compared to the baselines. Here, Best-FO denotes the performance of LV-Rep using full observations as inputs, providing a reference on how well an algorithm can achieve most in our tests.

|  | HalfCheetah | Humanoid | Walker | Ant | Hopper |
|---|---|---|---|---|---|
| **$\mu$LV-Rep** | **3596.2 $\pm$ 874.5** | **806.7 $\pm$ 120.7** | **1298.1$\pm$ 276.3** | **1621.4 $\pm$ 472.3** | **1096.4 $\pm$ 130.4** |
| Dreamer-v2 | 2863.8 $\pm$ 386 | 672.5 $\pm$ 36.6 | **1305.8 $\pm$ 234.2** | 1252.1 $\pm$ 284.2 | 758.3 $\pm$ 115.8 |
| SAC-MLP | 1612.0 $\pm$ 223 | 242.1 $\pm$ 43.6 | 736.5 $\pm$ 65.6 | **1612.0 $\pm$ 223** | 614.15 $\pm$ 67.6 |
| SLAC | **3012.4 $\pm$ 724.6** | 387.4 $\pm$ 69.2 | 536.5 $\pm$ 123.2 | 1134.8 $\pm$ 326.2 | 739.3 $\pm$ 98.2 |
| PSR | 2679.75$\pm$386 | 534.4 $\pm$ 36.6 | 862.4 $\pm$ 355.3 | 1128.3 $\pm$ 166.6 | 818.8 $\pm$ 87.2 |
| Best-FO | 5557.6$\pm$439.5 | 1086$\pm$278.2 | 2523.5$\pm$333.9 | 2511.8$\pm$460.0 | 2204.8$\pm$496.0 |
|  | Cheetah-run | Walker-run | Hopper-run | Humanoid-run | Pendulum |
| **$\mu$LV-Rep** | 525.3 $\pm$ 89.2 | **702.3 $\pm$ 124.3** | **69.3$\pm$ 12.8** | **9.8 $\pm$ 6.4** | 168.2 $\pm$ 5.3 |
| Dreamer-v2 | **602.3 $\pm$ 48.5** | 438.2 $\pm$ 78.2 | 59.2 $\pm$ 15.9 | 2.3 $\pm$ 0.4 | **172.3 $\pm$ 8.0** |
| SAC-MLP | 483.3 $\pm$ 77.2 | 279.8 $\pm$ 190.6 | 19.2 $\pm$ 2.3 | 1.2 $\pm$ 0.1 | 163.6 $\pm$ 9.3 |
| SLAC | 105.1 $\pm$ 30.1 | 139.2 $\pm$ 3.4 | 36.1 $\pm$ 15.3 | 0.9 $\pm$ 0.1 | **167.3 $\pm$ 11.2** |
| PSR | 173.7 $\pm$ 25.7 | 57.4 $\pm$ 7.4 | 23.2 $\pm$ 9.5 | 0.8 $\pm$ 0.1 | 159.4 $\pm$ 9.2 |
| Best-FO | 639.3$\pm$24.5 | 724.2$\pm$37.8 | 72.9$\pm$40.6 | 11.8$\pm$6.8 | 167.1$\pm$3.1 |

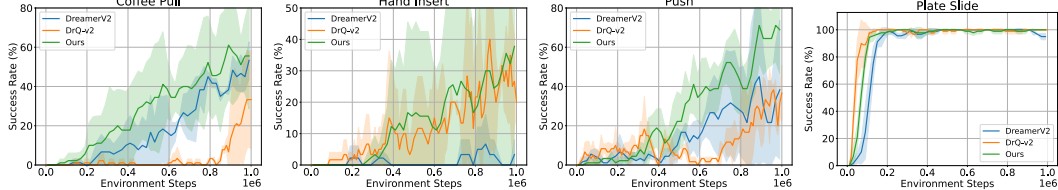

Figure 1: Learning curves on visual control tasks from DeepMind Control Suites measured by episodic return. Model-based methods outperform model-free method by a large margin on this domains. Our method demonstrates the best sample efficiency on most of the tasks.

Two baselines are used in this experiment: a model-based baseline Dreamer (Hafner et al., 2021), and DrQ-v2, a model-free baseline for visual continuous control (Yarats et al., 2021a). Figures 1 and 2 present the results. We compare all algorithms after running 500K environment steps on the DeepMind Control Suites and 1 million environment steps on Meta-world. All results are averaged across four random seeds. We observed that the proposed method achieves competitive or superior performance compared to both Dreamer-V2 and DrQ-v2 on all tested benchmarks, illustrating the versatility of our approach across a spectrum of complex visual control tasks.

Figure 2: Learning curves on visual robotic manipulation tasks from Meta-world measured by success rate. Our method shows better or comparable sample efficiency to model-free and model-based methods.

## 8 CONCLUSION

In this paper, we aimed to develop a practical RL algorithm for structured POMDPs that obtained efficiency in terms of both statistical and computational complexity. We revealed some of the challenges in computationally exploiting the low-rank structure of a POMDP, then derived a linear representation for the $Q^\pi$-function, which automatically implies a practical learning method, with tractable planning and exploration, as in $\mu$LV-Rep. We theoretically analyzed the sub-optimality of the proposed $\mu$LV-Rep, and empirically demonstrated its advantages on several benchmarks.

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
