## A   MORE RELATED WORK

**Partially Observable RL.**   The majority of existing practical RL algorithms for partially observable settings can also be categorized into model-based *vs*. model-free.

The model-based algorithms (Kaelbling et al., 1998) for partially observed scenarios are naturally derived based on the definition of POMDPs, where both the emission and transition models are learned from data. The planning procedure for optimal policy is conducted over the posterior of latent state, *i.e.*, beliefs, which is approximately inferred based on learned dynamics and emission model. With different model parametrizations, (ranging from Gaussian processes to deep models), and different planning methods, a family of algorithms has been proposed (Deisenroth & Peters, 2012; Igl et al., 2018; Gregor et al., 2019; Zhang et al., 2019; Lee et al., 2020; Hafner et al., 2021). However, due to the compounding errors from **i)**, mismatch in model parametrization, **ii)**, inaccurate beliefs calculation, **iii)**, approximation in planning over nonlinear dynamics, and **iv)**, neglecting of exploration, such methods might suffer from sub-optimal performances in practice.

As we discussed in Section 1, the memory-based policy and value function have been exploited to extend the MDP-based model-free RL algorithms to handle the non-Markovian dependency induced by partial observations. For example, the value-based algorithms introduces memory-based neural networks to Bellman recursion, including temporal difference learning with explicit concatenation of 4 consecutive frames as input (Mnih et al., 2013) or recurrent neural networks for longer windows (Bakker, 2001; Hausknecht & Stone, 2015; Zhu et al., 2017), and DICE (Nachum & Dai, 2020) with features extracted from transformer (Jiang et al., 2021); the policy gradient-based algorithms have been extended to partially observable setting by introducing recurrent neural network for policy parametrization (Schmidhuber, 1990; Wierstra et al., 2007; Heess et al., 2015; Ni et al., 2021). The actor-critic approaches exploits memory-based value and policy together (Ni et al., 2021; Meng et al., 2021). Despite their simplicity in the algorithm extension, these algorithms demonstrate potentials in real-world applications. However, the it has been observed that the sample complexity for purely model-free RL with partial observations is very high (Mnih et al., 2013; Barth-Maron et al., 2018; Yarats et al., 2021b), and the exploration remains difficult, and thus, largely neglected.

## B   OBSERVABILITY APPROXIMATION

Although the proposed $\mu$LV-Rep is designed based on the $L$-step decodability in POMDPs, Golowich et al. (2022) shows that the $\gamma$-observable POMDPs can be $\epsilon$-approximated with a $L = \tilde{O}\left(\gamma^{-4}\log\left(|S|/\epsilon\right)\right)$-step decodable POMDP. By exploiting the low-rank structure in the latent dynamics, this result has been extend with function approximator (Uehara et al., 2022). Specifically,

**Theorem 4** (Proprosition 7 (Guo et al., 2023), Lemma 12 (Uehara et al., 2022)). *Given a $\gamma$-observable POMDP with $d$-rank latent transition, there exists an $L$-step decodable POMDP $\mathcal{M}$ with $L = \tilde{O}\left(\gamma^{-4}\log\left(d/\epsilon\right)\right)$, $\forall \epsilon > 0$, such that*

$$\mathbb{E}_{a_{1:h},o_{2:h}\sim\pi}\left[\left\|\mathbb{P}_h\left(o_{h+1}|o_{1:h},a_{1:h}\right) - \mathbb{P}_h^{\mathcal{M}}\left(o_{h+1}|x_h,a_h\right)\right\|_1\right] \leqslant \epsilon. \tag{19}$$

*where $\pi_h \in \Delta\left(\prod_{h=1}^H \mathcal{A}^{\mathscr{H}_h}\right)$ with $\mathscr{H}_h := \mathcal{A}^{h-1} \times \mathcal{O}^h$, is mapping the whole history to a distribution of action.*

With this understanding, the proposed $\mu$LV-Rep can be directly applied for $\gamma$-observable POMDPs, while still maintains theoretical guarantees. Due to the space limitation, please refer to Uehara et al. (2022); Guo et al. (2023) for the details of the proofs.

## C   MOMENT MATCHING POLICY

We provide a formal definition of the moment matching policy here.

**Definition 5** (Moment Matching Policy (Efroni et al., 2022)). *With the $L$-decodability assumption, for $h \in [H]$, $h' \in [h - L + 1, h]$ and $l = h' - h + L - 1$, we can define the moment matching policy $\nu^{\pi,h} = \{\nu_{h'}^{h,\pi} : \mathcal{S}^l \times \mathcal{O}^l \times \mathcal{A}^{l-1} \to \Delta(\mathcal{A})\}_{h'=h-L+1}^h$ introduced by Efroni et al. (2022) , such that*

$$\nu_{h'}^{h,\pi}(a_{h'}|(s_{h-L+1:h'}, o_{h-L+1:h'}, a_{h-L+1:h'-1}))$$
$$:= \mathbb{E}_\pi^{\mathcal{P}}[\pi_{h'}(a_{h'}|x_{h'})|(s_{h-L+1:h'}, o_{h-L+1:h'}, a_{h-L+1:h'-1})], \quad \forall h' \leqslant h - 1,$$

---

**Algorithm 2** Offline Learning for $L$-step decodable POMDPs with Latent Variable Representation

---

1: **Input:** Model Class $\mathcal{M} = \{\{(p_h(z|x_h, a_h), p_h(o_{h+1}|z)\}_{h \in [H]}\}$, Variational Distribution Class $\mathcal{Q} = \{\{q_h(z|x_h, a_h, o_{h+1})\}_{h \in [H]}\}$, Offline Dataset $\{\mathcal{D}_h\}_{h=1}^H$

2: Learn the latent variable model $\hat{p}(z|x_h, a_h)$ with $\mathcal{D}_h$ via maximizing the ELBO, and obtain the learned model $\widehat{\mathcal{P}} = \{(\hat{p}_h(z|x_h, a_h), \hat{p}_h(o_{h+1}|z))\}_{h \in [H]}$.

3: Set the exploitation penalty $\hat{b}_h(s, a)$ with $\mathcal{D}_k$.

4: Learn the policy $\hat{\pi}^* = \arg\max_\pi V^\pi_{\widehat{\mathcal{P}}, r - \hat{b}_k}$.

5: **Return** $\hat{\pi}^*$.

---

and $\nu_h^{\pi, h} = \pi_h$. We further define $\tilde{\pi}^h$, which takes first $h - L$ actions from $\pi$ and the remaining $L$ actions from $\nu^{\pi, h}$.

The main motivation to define such moment matching policy is that, we want to define a policy that is conditionally independent from the past history for theoretical justification while indistinguishable from the history dependent policy to match the practical algorithm. By Lemma B.2 in Efroni et al. (2022), under the $L$-decodability assumption, for a fixed $h \in [H]$, we have $d_h^{\mathcal{P}, \pi}(x_h) = d_h^{\mathcal{P}, \tilde{\pi}_h}(x_h)$, for all $L$-step policy $\pi$ and $x_h \in \mathcal{X}_h$. As $\nu_h^{\pi, h} = \pi_h$. we have $d_h^{\mathcal{P}, \pi}(x_h, a_h) = d_h^{\mathcal{P}, \tilde{\pi}^h}(x_h, a_h)$, and hence $\mathbb{E}_\pi^{\mathcal{P}}(x_h, a_h) = \mathbb{E}_{\tilde{\pi}^h}^{\mathcal{P}}(x_h, a_h)$. This enables the factorization in (14) without the dependency of the overlap observation trajectory.

## D    PESSIMISTIM IN OFFLINE SETTING

Similar to Uehara et al. (2021); Ren et al. (2023b), the proposed algorithm can be directly extended to the offline setting by converting the optimism into the pessimism. Specifically, we can learn the latent variable model, set the penalty with the data and perform planning with the penalized reward. The whole algorithm is shown in Algorithm 2. Following the identical proof strategy from Uehara et al. (2021); Ren et al. (2023b), we can obtain a similar sub-optimal gap guarantee for $\hat{\pi}^*$.

## E    TECHNICAL BACKGROUND

In this section, we revisit several core concepts of the kernel and the reproducing kernel Hilbert space (RKHS) that will be used in the theoretical analysis. For a complete introduction, we refer the reader to Ren et al. (2023b).

**Definition 6** (Kernel and Reproducing Kernel Hilbert Space (RKHS) (Aronszajn, 1950; Paulsen & Raghupathi, 2016))**.** *The function $k : \mathcal{X} \times \mathcal{X} \to \mathbb{R}$ is called a kernel on $\mathcal{X}$ if there exists a Hilbert space $\mathcal{H}$ and a mapping $\phi : \mathcal{X} \to \mathcal{H}$ (termed as a feature map), such that $\forall x, x' \in \mathcal{X}$, $k(x, x') = \langle \phi(x), \phi(x') \rangle_{\mathcal{H}}$. The kernel $k$ is said to be positive semi-definite if $\forall n \geqslant 1$, $\{a_i\}_{i \in [n]} \subset \mathbb{R}$ and mutually distinct $\{x_i\}_{i \in [n]}$, we have*

$$\sum_{i \in [n]} \sum_{j \in [n]} a_i a_j k(x_i, x_j) \geqslant 0.$$

*The kernel $k$ is said to be positive definite if the inequality is strict (which means we can replace $\geqslant$ with $>$).*

*With a given kernel $k$, we can define the Hilbert space $\mathcal{H}_k$ consists of $\mathbb{R}$-valued function on $\mathcal{X}$ as a reproducing kernel Hilbert space associated with $k$ if both of the following conditions hold:*

- *$\forall x \in \mathcal{X}$, $k(x, \cdot) \in \mathcal{H}_k$.*

- *Reproducing Property: $\forall x \in \mathcal{X}$, $f \in \mathcal{H}_k$, $f(x) = \langle f, k(x, \cdot) \rangle_{\mathcal{H}_k}$.*

*The RKHS norm of $f \in \mathcal{H}_k$ is induced by the inner product, i.e. $\|f\|_{\mathcal{H}_k} := \sqrt{\langle f, f \rangle_{\mathcal{H}_k}}$.*

**Theorem 7** (Mercer's Theorem (Riesz & Nagy, 2012; Steinwart & Scovel, 2012))**.** *Let $k$ be a continuous positive definite kernel defined on $\mathcal{X} \times \mathcal{X}$. There exists at most countable $\{\mu_i\}_{i \in I}$ such that $\mu_1 \geqslant \mu_2 \geqslant \cdots > 0$ and a set of orthonormal basis $\{e_i\}_{i \in I}$ on $L_2(\mu)$ where $\mu$ is a Borel measure*

*on $\mathcal{X}$, such that*

$$\forall x, x' \in \mathcal{X}, \quad k(x, x') = \sum_{i \in I} \mu_i e_i(x) e_i(x'),$$

*where the convergence is absolute and uniform.*

**Definition 8** (Random Feature). *The kernel $k : \mathcal{X} \times \mathcal{X} \to \mathbb{R}$ has a random feature representation if there exists a function $\psi : \mathcal{X} \times \Xi \to \mathbb{R}$ and a probability measure $P$ over $\Xi$ such that*

$$k(x, x') = \int_\Xi \psi(x; \xi) \psi(x'; \xi) dP(\xi).$$

**Remark (random feature quadrature):** We here justify the random feature quadrature (Ren et al., 2023b) for completeness.

We can represent $Q_h^\pi$ as an expectation,

$$Q_h^\pi(x_h, a_h) = \langle p(z|x_h), w_h^\pi(z) \rangle = \mathbb{E}_{p(z|x_h)} [w_h^\pi(z)]_{L_2(\mu)}$$

Under the assumption that $w_h^\pi(\cdot) \in \mathcal{H}_k$, where $\mathcal{H}_k$ denoting some RKHS with some kernel $k(\cdot, \cdot)$. When $k(\cdot, \cdot)$ can be represented through random feature, *i.e.*,

$$k(x, y) = \mathbb{E}_{P(\xi)} [\psi(x; \xi) \psi(y; \xi)],$$

the $w_h^\pi(z)$ admits a representation as

$$w_h^\pi(z) = \mathbb{E}_{P(\xi)} [\tilde{w}_h^\pi(\xi) \psi(z; \xi)].$$

Therefore, we plug this random feature representation of $w_h^\pi(z)$ to $Q_h^\pi(x_h, a_h)$, we obtain

$$Q_h^\pi(x_h, a_h) = \mathbb{E}_{p(z|x_h), P(\xi)} [\tilde{w}_h^\pi(\xi) \psi(z; \xi)]. \tag{20}$$

Applying Monte-Carlo approximation to (20), we obtain the random feature quadrature in (18).

## F  THEORETICAL ANALYSIS

### F.1  TECHNICAL CONDITIONS

We adopt the following assumptions for the reproducing kernel, which have been used in Ren et al. (2023b) for the MDP setting.

**Assumption 3** (Regularity Conditions). *$\mathcal{Z}$ is a compact metric space with respect to the Lebesgue measure $\nu$ when $\mathcal{Z}$ is continuous. Furthermore, $\int_\mathcal{Z} k(z, z) d\nu \leqslant 1$.*

**Assumption 4** (Eigendecay Conditions). *Assume $\{\nu_i\}_{i \in I}$ defined in Theorem 7 satisfies one of the following conditions:*

- *$\beta$-finite spectrum: for some positive integer $\beta$, we have $\nu_i = 0$, $\forall i > \beta$.*

- *$\beta$-polynomial decay: $\nu_i \leqslant C_0 i^{-\beta}$ with absolute constant $C_0$ and $\beta > 1$.*

- *$\beta$-exponential decay: $\nu_i \leqslant C_1 \exp(-C_2 i^\beta)$, with absolute constants $C_1$, $C_2$ and $\beta > 0$.*

*We will use $C_{\mathrm{poly}}$ to denote constants in the analysis of $\beta$-polynomial decay that only depends on $C_0$ and $\beta$, and $C_{\mathrm{exp}}$ to denote constants in the analysis of $\beta$-exponential decay that only depends on $C_1$, $C_2$ and $\beta$, to simplify the dependency of the constant terms. Both of $C_{\mathrm{poly}}$ and $C_{\mathrm{exp}}$ can be varied step by step.*

### F.2  FORMAL PROOF

Before we proceed, we first define

$$\rho_{k,h} = \frac{1}{k} \sum_{i \in [K]} d_{\mathcal{P},h}^{\pi_k},$$

and $\circ^L \mathcal{U}(\mathcal{A})$ means uniformly taking actions in the consecutive $L$ steps.

**Lemma 9** (*L-step back inequality for the true model*). *Given a set of functions* $[g_h]_{h\in[H]}$, *where* $g_h : \mathcal{X} \times \mathcal{A} \to \mathbb{R}$, $\|g_h\|_\infty \leqslant B$, $\forall h \in [H]$, *we have that* $\forall \pi$,

$$\sum_{h\in[H]} \mathbb{E}_\pi^{\mathcal{P}}[g(x_h, a_h)] \leqslant \sum_{h\in[H]} \mathbb{E}_{(x_{h-L}, a_{h-L})\sim d_{\mathcal{P},h-L}^\pi}^{\mathcal{P}} \left[ \|p^*(\cdot|x_{h-L}, a_{h-L})\|_{L_2(\mu), \Sigma_{\rho_{k,h-L}, p^*}^{-1}} \right]$$
$$\cdot \sqrt{k|\mathcal{A}|^L \cdot \mathbb{E}_{(\tilde{x}_h, \tilde{a}_h)\sim\rho_{k,h-L}\circ^L\mathcal{U}(\mathcal{A})}[g(\tilde{x}_h, \tilde{a}_h)^2] + \lambda B^2 C}$$

*Proof.* The proof can be adapted from the proof of Lemma 6 in Ren et al. (2023b), and we include it for the completeness. Recall the moment matching policy $\nu^\pi$ Since $\nu^{\pi,h}$ does not depend on $(x_{h-L}, a_{h-L})$, we can make the following decomposition:

$$\mathbb{E}_{\tilde{\pi}^h}^{\mathcal{P}}(x_h, a_h)$$
$$= \mathbb{E}_{(x_{h-L}, a_{h-L})\sim\pi}^{\mathcal{P}} \left[ \int_{s_{h-L+1}} \langle p^*(\cdot|x_{h-L}, a_{h-L}), p^*(s_{h-L+1}|\cdot)\rangle_{L_2(\mu)} \cdot \mathbb{E}_{a_{h-L+1:h}\sim\nu^{\pi,h}}^{\mathcal{P}}[g(x_h, a_h)|s_{h-L+1}]ds_{h-L+1} \right]$$
$$\leqslant \mathbb{E}_{(x_{h-L}, a_{h-L})\sim\pi}^{\mathcal{P}} \|p^*(\cdot|x_{h-L}, a_{h-L})\|_{L_2(\mu), \Sigma_{\rho_{k,h-L}, p^*}^{-1}}$$
$$\cdot \left\| \int_{s_{h-L+1}} p^*(s_{h-L+1}|\cdot)\mathbb{E}[g(x_h, a_h)|s_{h-L+1}, \nu^{\pi,h}]ds_{h-L+1} \right\|_{L_2(\mu), \Sigma_{\rho_{k,h-L}, p^*}} .$$

Direct computation shows that

$$\left\| \int_{s_{h-L+1}} p^*(s_{h-L+1}|\cdot)\mathbb{E}^{\mathcal{P}}[g(x_h, a_h)|s_{h-L+1}, \nu^{\pi,h}]ds_{h-L+1} \right\|_{L_2(\mu), \Sigma_{\rho_{k,h-L}, p^*}}$$
$$= k\mathbb{E}_{(\tilde{x}_{h-L}, \tilde{a}_{h-L})\sim\rho_{k,h-L}} \left[ \mathbb{E}_{s_{h-L+1}\sim\mathbb{P}_{h-L}^{\mathcal{P}}(\cdot|x_{h-L}, a_{h-L})}^{\mathcal{P}}[g(x_h, a_h)|s_{h-L+1}, \nu^{\pi,h}] \right]^2$$
$$+ \left\| \int_{s_{h-L+1}} p^*(s_{h-L+1}|\cdot) \cdot \mathbb{E}^{\mathcal{P}}[g(x_h, a_h)|s_{h-L+1}, \nu^{\pi,h}]ds_{h-L+1} \right\|_{\mathcal{H}}^2$$
$$\leqslant k\mathbb{E}_{(\tilde{x}_{h-L}, \tilde{a}_{h-L})\sim\rho_{k,h-L}} \mathbb{E}_{s_{h-L+1}\sim\mathbb{P}_{h-L}^{\mathcal{P}}(\cdot|x_{h-L}, a_{h-L}), a_{h-L+1:h}\sim\nu^{\pi,h}}^{\mathcal{P}}[g(x_h, a_h)]^2 + \lambda B^2 C$$
$$\leqslant k|\mathcal{A}|^L \mathbb{E}_{(\tilde{x}_h, \tilde{a}_h)\sim\rho_{k,h-L}\circ^L\mathcal{U}(\mathcal{A})}^{\mathcal{P}}[g(\tilde{x}_h, \tilde{a}_h)]^2 + \lambda B^2 C,$$

which finishes the proof. $\square$

**Lemma 10** (*L-step back inequality for the learned model*). *Assume we have a set of functions* $[g_h]_{h\in[H]}$, *where* $g_h : \mathcal{X} \times \mathcal{A} \to \mathbb{R}$, $\|g_h\|_\infty \leqslant B$, $\forall h \in [H]$. *Given Lemma 15, we have that* $\forall \pi$,

$$\sum_{h\in[H]} \mathbb{E}_\pi^{\widehat{\mathcal{P}}_k}[g(x_h, a_h)] \leqslant \sum_{h\in[H]} \mathbb{E}_{(x_{h-L}, a_{h-L})\sim d_{\widehat{\mathcal{P}}_k,h-L}^\pi}^{\widehat{\mathcal{P}}_k} \left[ \|\hat{p}(\cdot|x_{h-L}, a_{h-L})\|_{L_2(\mu), \Sigma_{\rho_{k,h-2L}\circ^L\mathcal{U}(\mathcal{A}), \hat{p}}^{-1}} \right]$$
$$\cdot \sqrt{k|\mathcal{A}|^L \cdot \mathbb{E}_{(\tilde{x}_h, \tilde{a}_h)\sim\rho_{k,h-2L}\circ^{2L}\mathcal{U}(\mathcal{A})}[g(\tilde{x}_h, \tilde{a}_h)^2] + \lambda B^2 C + kL|\mathcal{A}|^{L-1}B^2\zeta_k}$$

*Proof.* The proof can be adapted from the proof of Lemma 5 in Ren et al. (2023b), and we include it for the completeness. We define a similar moment matching policy and make the following decomposition:

$$\mathbb{E}_\pi^{\widehat{\mathcal{P}}_k}(x_h, a_h)$$
$$= \mathbb{E}_{(x_{h-L}, a_{h-L})\sim\pi}^{\widehat{\mathcal{P}}_k} \left[ \int_{s_{h-L+1}} \langle \hat{p}(\cdot|x_{h-L}, a_{h-L}), \hat{p}(s_{h-L+1}|\cdot)\rangle_{L_2(\mu)} \cdot \mathbb{E}^{\widehat{\mathcal{P}}_k}[g(x_h, a_h)|s_{h-L+1}, \nu^{\pi,h}]ds_{h-L+1} \right]$$
$$\leqslant \mathbb{E}_{(x_{h-L}, a_{h-L})\sim\pi}^{\widehat{\mathcal{P}}_k} \|\hat{p}(\cdot|x_{h-L}, a_{h-L})\|_{L_2(\mu), \Sigma_{\rho_{k,h-2L}\circ^L\mathcal{U}(\mathcal{A}), \hat{p}}^{-1}}$$
$$\cdot \left\| \int_{s_{h-L+1}} \hat{p}(s_{h-L+1}|\cdot)\mathbb{E}^{\widehat{\mathcal{P}}_k}[g(x_h, a_h)|s_{h-L+1}, \nu^{\pi,h}]ds_{h-L+1} \right\|_{L_2(\mu), \Sigma_{\rho_{k,h-2L}\circ^L\mathcal{U}(\mathcal{A}), \hat{p}}} .$$

Direct computation shows that

$$\left\| \int_{s_{h-L+1}} \hat{p}(s_{h-L+1}|\cdot) \mathbb{E}^{\widehat{\mathcal{P}}_k}[g(x_h, a_h)|s_{h-L+1}, \nu^{\pi,h}] ds_{h-L+1} \right\|^2_{L_2(\mu), \Sigma_{\rho_{k,h-2L} \circ^L \mathcal{U}(\mathcal{A}), \hat{p}}}$$

$$= k \mathbb{E}_{(\tilde{x}_{h-L}, \tilde{a}_{h-L}) \sim \rho_{k,h-2L} \circ^L \mathcal{U}(\mathcal{A})} \left[ \mathbb{E}_{s_{h-L+1} \sim \mathbb{P}^{\widehat{\mathcal{P}}_k}_{h-L}(\cdot|\tilde{x}_{h-L}, \tilde{a}_{h-L})} \mathbb{E}^{\widehat{\mathcal{P}}_k}[g(x_h, a_h)|s_{h-L+1}, \nu^{\pi,h}] \right]^2$$

$$+ \left\| \int_{s_{h-L+1}} \hat{p}(s_{h-L+1}|\cdot) \mathbb{E}[g(x_h, a_h)|s_{h-L+1}, \nu^{\pi,h}] ds_{h-L+1} \right\|^2_{\mathcal{H}}$$

$$\leqslant k \mathbb{E}_{(\tilde{x}_{h-L}, \tilde{a}_{h-L}) \sim \rho_{k,h-2L} \circ^L \mathcal{U}(\mathcal{A})} \mathbb{E}^{\widehat{\mathcal{P}}_k}_{s_{h-L+1} \sim \mathbb{P}^{\widehat{\mathcal{P}}_k}_{h-L}(\cdot|\tilde{x}_{h-L}, \tilde{a}_{h-L}), \nu^{\pi,h}}[g(x_h, a_h)]^2 + \lambda B^2 C$$

$$\leqslant k |\mathcal{A}|^L \mathbb{E}_{(\tilde{x}_{h-L}, \tilde{a}_{h-L}) \sim \rho_{k,h-2L} \circ^L \mathcal{U}(\mathcal{A})} \mathbb{E}^{\widehat{\mathcal{P}}_k}_{a_{h-L+1:h} \sim \circ^L \mathcal{U}(\mathcal{A})}[g(x_h, a_h)]^2 + \lambda B^2 C$$

$$\leqslant k |\mathcal{A}|^L \mathbb{E}_{(\tilde{x}_h, \tilde{a}_h) \sim \rho_{k,h-2L} \circ^{2L} \mathcal{U}(\mathcal{A})}[g(\tilde{x}_h, \tilde{a}_h)]^2 + kL|\mathcal{A}|^{L-1} B^2 \zeta_k + \lambda B^2 C,$$

where we use the MLE guarantee for each individual step to obtain the last inequality. This finishes the proof. $\qquad \square$

**Lemma 11** (Almost Optimism). *For episode $k \in [K]$, set*
$$\hat{b}_{k,h} = \min \left\{ \alpha_k \|\hat{p}_k(\cdot|x_{h-L}, a_{h-L})\|_{L_2(\mu), \hat{\Sigma}^{-1}_{k,h,\hat{p}_k}}, 2 \right\},$$
*with $\alpha_k = \frac{\sqrt{5kL|\mathcal{A}|^L \zeta_k + 4\lambda d}}{c}$,*
$$\hat{\Sigma}_{k,h,\hat{p}_k} : L_2(\mu) \to L_2(\mu), \quad \hat{\Sigma}_{k,h,\hat{p}_k} := \sum_{(x_{h,i}, a_{h,i}) \in \mathcal{D}_{k,h}} \left[ \hat{p}_k(z|x_{h,i}, a_{h,i}) \hat{p}_k(z|x_{h,i}, a_{h,i})^\top \right] + \lambda T_K^{-1}$$
*where $T_K$ is the integral operator associated with $K$ (i.e. $T_K f = \int f(x) K(x, \cdot) dx$) and $\lambda$ is set for different eigendecay of $K$ as follows:*

- *$\beta$-finite spectrum: $\lambda = \Theta(\beta \log K + \log(K|\mathcal{P}|/\delta))$*

- *$\beta$-polynomial decay: $\lambda = \Theta(C_{\text{poly}} K^{1/(1+\beta)} + \log(K|\mathcal{P}|/\delta))$;*

- *$\beta$-exponential decay: $\lambda = \Theta(C_{\exp}(\log K)^{1/\beta} + \log(K|\mathcal{P}|/\delta))$;*

*$c$ is an absolute constant, then with probability at least $1 - \delta$, $\forall k \in [K]$ we have*
$$V^{\pi^*, \widehat{\mathcal{P}}_k, r + \hat{b}_k} - V^{\pi^*, \mathcal{P}, r} \geqslant -\sqrt{|\mathcal{A}|^{L+1} \zeta_k}$$

*Proof.* With Lemma 14, we have that
$$V^{\pi^*, \widehat{\mathcal{P}}_k, r + \hat{b}^k} - V^{\pi^*, \mathcal{P}, r}$$

$$= \sum_{h \in [H]} \mathbb{E}_{(x_h, a_h) \sim d^{\pi^*}_{\widehat{\mathcal{P}}_k, h}} \left[ \hat{b}^k_h(x_h, a_h) + \mathbb{E}_{o' \sim \mathbb{P}^{\widehat{\mathcal{P}}_k}_h(\cdot|x_h, a_h)}[V^{\pi^*, \mathcal{P}, r}_{h+1}(x'_{h+1})] - \mathbb{E}_{o' \sim \mathbb{P}^{\mathcal{P}}_h(\cdot|x_h, a_h)}[V^{\pi^*, \mathcal{P}, r}_{h+1}(x'_{h+1})] \right]$$

$$\geqslant \sum_{h \in [H]} \mathbb{E}_{(x_h, a_h) \sim d^{\pi^*}_{\widehat{\mathcal{P}}_k, h}} \left[ \min \left[ c\alpha_k \|\hat{p}(\cdot|x_{h-L}, a_{h-L})\|_{L_2(\mu), \Sigma^{-1}_{\rho_{k,h-L}, \hat{p}}}, 2 \right] + \mathbb{E}_{o' \sim \mathbb{P}^{\widehat{\mathcal{P}}_k}_h(\cdot|x_h, a_h)}[V^{\pi^*, \mathcal{P}, r}_{h+1}(x'_{h+1})] \right.$$

$$\left. - \mathbb{E}_{o' \sim \mathbb{P}^{\mathcal{P}}_h(\cdot|x_h, a_h)}[V^{\pi^*, \mathcal{P}, r}_{h+1}(x'_{h+1})] \right],$$

where in the last step we replace the empirical covariance with the population counterpart thanks to Lemma 17 in Ren et al. (2023b). Define
$$g_h(z_h, a_h) = \mathbb{E}_{o' \sim \mathbb{P}^{\mathcal{P}}_h(\cdot|x_h, a_h)}[V^{\pi^*, \mathcal{P}, r}_{h+1}(x'_{h+1})] - \mathbb{E}_{o' \sim \mathbb{P}^{\widehat{\mathcal{P}}_k}_h(\cdot|x_h, a_h)}[V^{\pi^*, \mathcal{P}, r}_{h+1}(x'_{h+1})],$$
With Hölder's inequality, we have that $\|g_h\|_\infty \leqslant 2$.

Furthermore, with Lemma 10, we have that

$$\sum_{h \in [H]} \mathbb{E}_{(x_h, a_h) \sim d^{\pi^*}_{\widehat{\mathcal{P}}_k, h}} [g_h(x_h, a_h)]$$

$$\leqslant \sum_{h \in [H]} \mathbb{E}_{(x_{h-L}, a_{h-L}) \sim d^{\pi^*}_{\widehat{\mathcal{P}}_k, h}} \left[ \|\hat{p}(\cdot|x_{h-L}, a_{h-L})\|_{L_2(\mu), \Sigma^{-1}_{\rho_{k,h-L}, \hat{p}}} \right]$$

$$\cdot \sqrt{k|\mathcal{A}|^L \cdot \mathbb{E}_{(\tilde{x}_h, \tilde{a}_h) \sim \rho_{h-2L} \circ^{2L} \mathcal{U}(\mathcal{A})} [g(\tilde{x}_h, \tilde{a}_h)^2] + 4\lambda C + 4kL|\mathcal{A}|^{L-1}\zeta_k}$$

$$\leqslant \sum_{h \in [H]} \mathbb{E}_{(x_{h-L}, a_{h-L}) \sim d^{\pi^*}_{\widehat{\mathcal{P}}_k, h}} \left[ c\alpha_k \|\hat{p}(\cdot|x_{h-L}, a_{h-L})\|_{L_2(\mu), \Sigma^{-1}_{\rho_{k,h-L}, \hat{p}}} \right],$$

where we use Lemma 15 in the last step. Now we deal with the case with $h \in [L]$. Note that, $\forall h \in [L]$

$$\mathbb{E}_{(x_h, a_h) \sim \pi}[g_h(x_h, a_h)]$$

$$\leqslant |\mathcal{A}|^h \mathbb{E}_{x_1 \sim d_1, a_{1:h} \sim \circ^h \mathcal{U}(\mathcal{A})} \left\| \mathbb{P}^{\widehat{\mathcal{P}}_k}_h(\cdot|x_h, a_h) - \mathbb{P}^{\mathcal{P}}_h(\cdot|x_h, a_h) \right\|_1$$

$$\leqslant \sqrt{\mathbb{E}_{x_1 \sim d_1, a_{1:h} \circ^L \mathcal{U}(\mathcal{A})} \left\| \mathbb{P}^{\widehat{\mathcal{P}}_k}_h(\cdot|x_h, a_h) - \mathbb{P}^{\mathcal{P}}_h(\cdot|x_h, a_h) \right\|_1^2}$$

$$\leqslant \sqrt{|\mathcal{A}|^h \zeta_k},$$

where in the last step we use Lemma 15. We finish the proof by summing over $h \in [L]$. $\qquad \square$

**Lemma 12** (Regret). *With probability at least $1 - \delta$, we have that*

- *For $\beta$-finite spectrum, we have*
$$\sum_{k=1}^K V^{\pi^*, \mathcal{P}, r} - V^{\pi_k, \mathcal{P}, r} \lesssim \sum_{k=1}^K V^{\pi^*, \mathcal{P}, r} - V^{\pi_k, \mathcal{P}, r} \lesssim H^2 \beta^{3/2} |\mathcal{A}|^L \log K \sqrt{CLK \log(K|\mathcal{M}|/\delta)};$$

- *For $\beta$-polynomial decay, we have*
$$\sum_{k=1}^K V^{\pi^*, \mathcal{P}, r} - V^{\pi_k, \mathcal{P}, r} \lesssim C_{\text{poly}} H^2 |\mathcal{A}|^L K^{\frac{1}{2} + \frac{1}{1+\beta}} \sqrt{CL \log(K|\mathcal{M}|/\delta)};$$

- *For $\beta$-exponential decay, we have*
$$\sum_{k=1}^K V^{\pi^*, \mathcal{P}, r} - V^{\pi_k, \mathcal{P}, r} \lesssim C_{\text{exp}} H^2 |\mathcal{A}|^L (\log K)^{1 + \frac{3}{2\beta}} \sqrt{CLK \log(K|\mathcal{M}|/\delta)};$$

*Proof.* With Lemma 11 and Lemma 14, we have

$$V^{\pi^*, \mathcal{P}, r} - V^{\pi_k, \mathcal{P}, r}$$

$$\leqslant V^{\pi^*, \widehat{\mathcal{P}}_k, r + \widehat{b}^k} + \sqrt{|\mathcal{A}|^{L+1} \zeta_k} - V^{\pi_k, \mathcal{P}, r}$$

$$\leqslant V^{\pi^k, \widehat{\mathcal{P}}_k, r + \widehat{b}^k} + \sqrt{|\mathcal{A}|^{L+1} \zeta_k} - V^{\pi_k, \mathcal{P}, r}$$

$$= \sum_{h \in [H]} \mathbb{E}_{(x_h, a_h) \sim d^{\pi_k}_{\mathcal{P}, h}} \left[ \widehat{b}^k_h(x_h, a_h) + \mathbb{E}_{o' \sim \mathbb{P}^{\widehat{\mathcal{P}}_k}_h(\cdot|x_h, a_h)} \left[ V^{\pi_k, \widehat{\mathcal{P}}_k, r + \widehat{b}^k_h}_{h+1}(x'_{h+1}) \right] - \mathbb{E}_{o' \sim \mathbb{P}^{\mathcal{P}}_h(\cdot|x_h, a_h)} \left[ V^{\pi_k, \widehat{\mathcal{P}}_k, r + \widehat{b}^k_h}_{h+1}(x'_{h+1}) \right] \right],$$

$$+ \sqrt{|\mathcal{A}|^{L+1} \zeta_k}.$$

Note that $\left\| \widehat{b}^k_h \right\|_\infty \leqslant 2$. Applying Lemma 9, we have that

$$\sum_{h \in [H]} \mathbb{E}_{(x_h, a_h) \sim d^{\pi_k}_{\mathcal{P}, h}} \left[ \widehat{b}^k_h(x_h, a_h) \right]$$

$$\leqslant \sum_{h \in [H]} \mathbb{E}_{(\tilde{x}_{h-L}, \tilde{a}_{h-L}) \sim d^{\pi_k}_{\mathcal{P}, h}} \left[ \|p^*(\cdot|x_{h-L}, a_{h-L})\|_{L_2(\mu), \Sigma^{-1}_{\rho_{k,h-L}, p^*}} \right]$$

$$\cdot \sqrt{k|\mathcal{A}|^L \cdot \mathbb{E}_{(\tilde{x}_h, \tilde{a}_h) \sim \rho_{k,h-L} \circ^L \mathcal{U}(\mathcal{A})} \left[ \widehat{b}^k_h(\tilde{x}_h, \tilde{a}_h)^2 \right] + 4\lambda C}$$

Following the proof of Lemma 8 in Ren et al. (2023b), we have that:

- for $\beta$-finite spectrum,
$$k\mathbb{E}_{(\tilde{x}_h,\tilde{a}_h)\sim\rho_{k,h-L}\circ^L\mathcal{U}(\mathcal{A})}\left[\widehat{b}_h^k(\tilde{x}_h,\tilde{a}_h)^2\right] = O(\beta\log K);$$

- for $\beta$-polynomial decay,
$$k\mathbb{E}_{(\tilde{x}_h,\tilde{a}_h)\sim\rho_{k,h-L}\circ^L\mathcal{U}(\mathcal{A})}\left[\widehat{b}_h^k(\tilde{x}_h,\tilde{a}_h)^2\right] = O\left(C_{\text{poly}}K^{\frac{1}{2(1+\beta)}}\log K\right);$$

- for $\beta$-exponential decay,
$$k\mathbb{E}_{(\tilde{x}_h,\tilde{a}_h)\sim\rho_{k,h-L}\circ^L\mathcal{U}(\mathcal{A})}\left[\widehat{b}_h^k(\tilde{x}_h,\tilde{a}_h)^2\right] = O\left(C_{\exp}(\log K)^{1+1/\beta}\right).$$

We then consider
$$\sum_{h\in[H]}\mathbb{E}_{(x_h,a_h)\sim d_{\mathcal{P},h}^{\pi_k}}\left[\mathbb{E}_{o'\sim\widehat{\mathbb{P}}_h^k(\cdot|x_h,a_h)}\left[V_{h+1}^{\pi_k,\widehat{\mathcal{P}}_k,r+\widehat{b}_h^k}(x'_{h+1})\right] - \mathbb{E}_{o'\sim\mathbb{P}_h^{\mathcal{P}}(\cdot|x_h,a_h)}\left[V_{h+1}^{\pi_k,\widehat{\mathcal{P}}_k,r+\widehat{b}_h^k}(x'_{h+1})\right]\right].$$
Define
$$g(x_h,a_h) = \frac{1}{2H+1}\left[\mathbb{E}_{o'\sim\widehat{\mathbb{P}}_h^k(\cdot|x_h,a_h)}\left[V_{h+1}^{\pi_k,\widehat{\mathcal{P}}_k,r+\widehat{b}_h^k}(x'_{h+1})\right] - \mathbb{E}_{o'\sim\mathbb{P}_h^{\mathcal{P}}(\cdot|x_h,a_h)}\left[V_{h+1}^{\pi_k,\widehat{\mathcal{P}}_k,r+\widehat{b}_h^k}(x'_{h+1})\right]\right].$$
With Hölder's inequality and note that $\left\|\widehat{b}_h^k\right\| \leqslant 2$, we have that $\|g\|_\infty \leqslant 2$. With Lemma 9, we have that
$$\sum_{h\in[H]}\mathbb{E}_{(x_h,a_h)\sim d_{\mathcal{P},h}^{\pi_k}}[g(x_h,a_h)]$$
$$\leqslant \sum_{h\in[H]}\mathbb{E}_{(x_{h-L},a_{h-L})\sim d_{\mathcal{P},h}^{\pi_k}}^{\mathcal{P}}\left[\|p^*(\cdot|x_{h-L},a_{h-L})\|_{L_2(\mu),\Sigma_{\rho_{k,h-L},p^*}^{-1}}\right]\cdot\sqrt{k|\mathcal{A}|^L\mathbb{E}_{(\tilde{x}_h,\tilde{a}_h)\sim\rho_{k,h-L}\circ^L\mathcal{U}(\mathcal{A})}[g(\tilde{x}_h,\tilde{a}_h)^2] + 4\lambda C}$$
$$\leqslant \sum_{h\in[H]}\mathbb{E}_{(x_{h-L},a_{h-L})\sim d_{\mathcal{P},h}^{\pi_k}}^{\mathcal{P}}\left[\|p^*(\cdot|x_{h-L},a_{h-L})\|_{L_2(\mu),\Sigma_{\rho_{k,h-L},p^*}^{-1}}\right]\cdot\sqrt{k|\mathcal{A}|^L\zeta_k + 4\lambda C}$$
$$\leqslant c\alpha_k\sum_{h\in[H]}\mathbb{E}_{(x_{h-L},a_{h-L})\sim d_{\mathcal{P},h}^{\pi}}^{\mathcal{P}}\left[\|p^*(\cdot|x_{h-L},a_{h-L})\|_{L_2(\mu),\Sigma_{\rho_{k,h-L},p^*}^{-1}}\right]$$
With Cauchy-Schwartz inequality, we know that
$$\sum_{k\in[K]}\mathbb{E}_{(x_{h-L},a_{h-L})\sim d_{\mathcal{P},h}^{\pi_k}}^{\mathcal{P}}\left[\|p^*(\cdot|x_{h-L},a_{h-L})\|_{L_2(\mu),\Sigma_{\rho_{k,h-L},p^*}^{-1}}\right]$$
$$\leqslant\sqrt{K\sum_{k\in[K]}\mathbb{E}_{(x_{h-L},a_{h-L})\sim d_{\mathcal{P},h}^{\pi_k}}^{\mathcal{P}}\left[\|p^*(\cdot|x_{h-L},a_{h-L})\|_{L_2(\mu),\Sigma_{\rho_{k,h-L},p^*}^{-1}}^2\right]}.$$
Following the proof of Lemma 8 in Ren et al. (2023b), we have that

- for $\beta$-finite spectrum,
$$\sum_{k\in[K]}\mathbb{E}_{(x_{h-L},a_{h-L})\sim d_{\mathcal{P},h}^{\pi_k}}^{\mathcal{P}}\left[\|p^*(\cdot|x_{h-L},a_{h-L})\|_{L_2(\mu),\Sigma_{\rho_{k,h-L},p^*}^{-1}}^2\right] = O(\beta\log K);$$

- for $\beta$-polynomial decay,
$$\sum_{k\in[K]}\mathbb{E}_{(x_{h-L},a_{h-L})\sim d_{\mathcal{P},h}^{\pi_k}}^{\mathcal{P}}\left[\|p^*(\cdot|x_{h-L},a_{h-L})\|_{L_2(\mu),\Sigma_{\rho_{k,h-L},p^*}^{-1}}^2\right] = O\left(C_{\text{poly}}K^{\frac{1}{2(1+\beta)}}\log K\right);$$

- for $\beta$-exponential decay,
$$\sum_{k\in[K]}\mathbb{E}_{(x_{h-L},a_{h-L})\sim d_{\mathcal{P},h}^{\pi_k}}^{\mathcal{P}}\left[\|p^*(\cdot|x_{h-L},a_{h-L})\|_{L_2(\mu),\Sigma_{\rho_{k,h-L},p^*}^{-1}}^2\right] = O\left(C_{\exp}(\log K)^{1+1/\beta}\right).$$

Combine the previous steps and take the dominating term out, we have that

- for $\beta$-finite spectrum,
$$\sum_{k=1}^{K} V^{\pi^*,\mathcal{P},r} - V^{\pi_k,\mathcal{P},r} \lesssim H^2 \beta^{3/2} |\mathcal{A}|^L \log K \sqrt{CLK \log(K|\mathcal{M}|/\delta)};$$

- for $\beta$-polynomial decay,
$$\sum_{k=1}^{K} V^{\pi^*,\mathcal{P},r} - V^{\pi_k,\mathcal{P},r} \lesssim C_{\text{poly}} H^2 |\mathcal{A}|^L K^{\frac{1}{2}+\frac{1}{1+\beta}} \sqrt{CL \log(K|\mathcal{M}|/\delta)};$$

- for $\beta$-exponential decay,
$$\sum_{k=1}^{K} V^{\pi^*,\mathcal{P},r} - V^{\pi_k,\mathcal{P},r} \lesssim C_{\text{exp}} H^2 |\mathcal{A}|^L (\log K)^{1+\frac{3}{2\beta}} \sqrt{CLK \log(K|\mathcal{M}|/\delta)};$$

which finishes the proof. $\qquad\square$

**Theorem 13** (PAC Guarantee). *After interacting with the environments for $KH$ episodes*

- $K = \Theta\left( \frac{CH^4 L \beta^3 |\mathcal{A}|^{2L} \log(|\mathcal{P}|/\delta)}{\varepsilon^2} \log^3\left( \frac{CH^4 L \beta^3 |\mathcal{A}|^{2L} \log(|\mathcal{P}|/\delta)}{\varepsilon^2} \right) \right)$ *for $\beta$-finite spectrum;*

- $K = \Theta\left( C_{\text{poly}} \left( \frac{H^2 L |\mathcal{A}|^L \sqrt{C \log(|\mathcal{P}|/\delta)}}{\varepsilon} \log^{3/2}\left( \frac{\sqrt{C} H^2 L |\mathcal{A}|^L \log(|\mathcal{P}|/\delta)}{\varepsilon} \right) \right)^{\frac{2(1+\beta)}{\beta-1}} \right)$ *for $\beta$-polynomial decay;*

- $K = \Theta\left( \frac{C_{\text{exp}} C H^4 L |\mathcal{A}|^{2L} \log(|\mathcal{P}|/\delta)}{\varepsilon^2} \log^{\frac{3+2\beta}{\beta}}\left( \frac{CH^4 L |\mathcal{A}|^{2L} \log(|\mathcal{P}|/\delta)}{\varepsilon^2} \right) \right)$ *for $\beta$-exponential decay;*

*we can obtain an $\varepsilon$-optimal policy with high probability.*

*Proof.* This is a direct extension of the proof of Theorem 9 in Ren et al. (2023b). $\qquad\square$

## G   TECHNICAL LEMMA

**Lemma 14** (Simulation Lemma). *For two MDPs $\mathcal{M} = (P, r)$ and $\mathcal{M}' = (P', r+b)$, we have*
$$V_{P',r+b}^{\pi} - V_{P,r}^{\pi}$$
$$= \sum_{h \in [H]} \mathbb{E}_{(s_h, a_h) \sim d_{P,\pi}^h} \left[ b_h(s_h, a_h) + \mathbb{E}_{s_{h+1} \sim P'(s_h, a_h)} \left[ V_{P',r+b,h+1}^{\pi}(s_{h+1}) \right] - \mathbb{E}_{s_{h+1} \sim P(s_h, a_h)} \left[ V_{P',r+b,h+1}^{\pi}(s_{h+1}) \right] \right],$$
*and*
$$V_{P',r+b}^{\pi} - V_{P,r}^{\pi}$$
$$= \sum_{h \in [H]} \mathbb{E}_{(s_h, a_h) \sim d_{P',\pi}^h} \left[ b_h(s_h, a_h) + \mathbb{E}_{s_{h+1} \sim P'(s_h, a_h)} \left[ V_{P,r,h+1}^{\pi}(s_{h+1}) \right] - \mathbb{E}_{s_{h+1} \sim P(s_h, a_h)} \left[ V_{P,r,h+1}^{\pi}(s_{h+1}) \right] \right],$$

For the proof, see Uehara et al. (2021) for an example.

**Lemma 15** (MLE Guarantee). *For any episoode $k \in [K]$, step $h \in [H]$, define $\rho_h$ as the joint distribution of $(x_h, a_h)$ in the dataset $\mathcal{D}_{h,k}$ at episode $k$. Then with probability at least $1 - \delta$, we have that*
$$\mathbb{E}_{(x_h,a_h)\sim\mathcal{D}_{h,k}} \left\| \mathbb{P}_h^{\mathcal{P}}(\cdot|x_h, a_h) - \mathbb{P}_h^{\widehat{\mathcal{P}}_k}(\cdot|x_h, a_h) \right\|_1^2 \leqslant \zeta_k,$$
*where $\zeta_k = O(\log(Hk|\mathcal{M}|/\delta)/k)$*

For the proof, see Agarwal et al. (2020).

# H    IMPLEMENTATION DETAILS ON IMAGE-BASED CONTINUOUS CONTROL

We evaluate our method on DeepMind Control Suites (Tassa et al., 2018) [1] and Meta-world (Yu et al., 2019) [2] to demonstrate its capability for complex visual control tasks. Meta-world is an open-source simulated benchmark consisting of 50 distinct robotic manipulation tasks. The DeepMind Control Suite is a set of continuous control tasks with a standardized structure and interpretable rewards, intended to serve as performance benchmarks for reinforcement learning agents. The visualization of the two domains is shown in Figures 3 and 4. With only one frame of the visual observation, we will miss some information related to the task, for example the speed, thus these tasks are partially observable.

In particular, we employ visual observations with dimensions of $64 \times 64 \times 3$ and apply a masked autoencoder (MAE) with a masking ratio of $0.75$ to learn representations for these visual observations (He et al., 2022). The MAE is first pre-trained with random trajectories at the beginning and then fine-tuned during the online learning procedure. It produces compact vector representations for the images, which are then forwarded as input to our representation learning method. A Recurrent State-Space Model (RSSM) (Hafner et al., 2019) forms the dynamics of the world model. The RSSM uses a sequence of deterministic recurrent states, from which it computes two distributions over stochastic states at each step. The posterior latent state incorporates information about the current representation. The learning goal is to use the prior latent state to predict the posterior without access to the current representation. We incorporate multi-step prediction in RSSM with one starting state and a sequence of actions. Since the representation is learned from MAE, we reconstruct the representation after we get the representation predicted by RSSM. We apply actor-critic Learning based on the representation learned by MAE and the dynamics learned by RSSM. The configuration of used tasks are given in Table 2. The hyperparameters used in MAE, RSSM and the RL agent are shown in Tables 3 and 4.

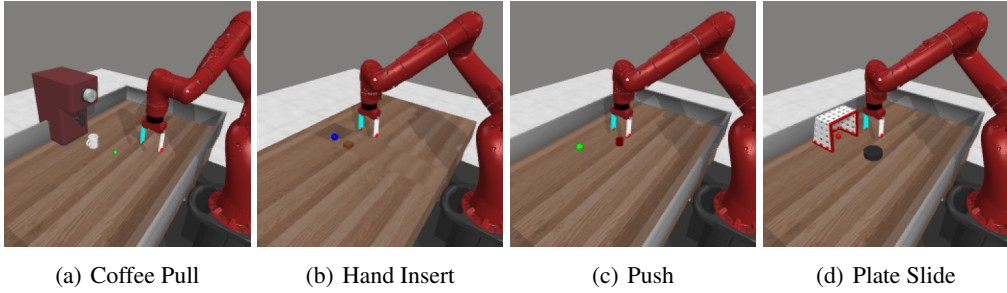

| (a) Coffee Pull | (b) Hand Insert | (c) Push | (d) Plate Slide |

Figure 3: Visualization of the visual robotic manipulation tasks in Meta-world.

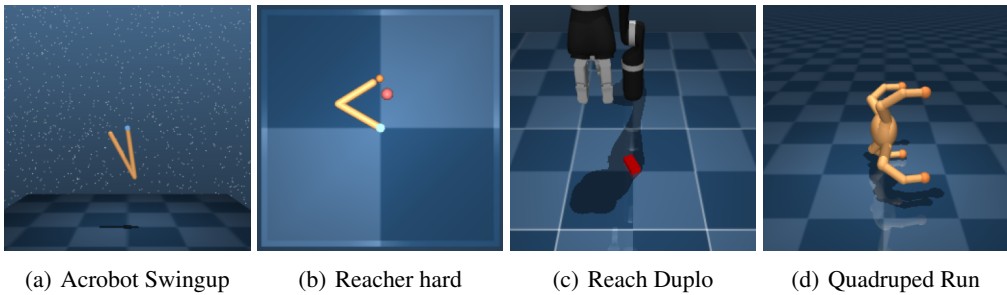

| (a) Acrobot Swingup | (b) Reacher hard | (c) Reach Duplo | (d) Quadruped Run |

Figure 4: Visualization of the visual control tasks in DeepMind Control Suites.

---

[1] https://github.com/google-deepmind/dm_control
[2] https://github.com/Farama-Foundation/Metaworld

Table 2: Configuration of environments.

| Hyperparameter | Value |
| --- | --- |
| Image observation | $64 \times 64 \times 3$ |
| Image normalization | Mean: $(0.485, 0.456, 0.406)$, Std: $(0.229, 0.224, 0.225)$ |
| Action repeat | 2 |
| Episode length | 500 (Meta-world), 1000 (DMC) |
| Normalize action | [-1,1] |
| Camera | corner2 (Meta-world), camera2 (DMC) |
| Total steps in environment | 1M (Meta-world), 0.5M (DMC) |

Table 3: Hyperparameters in world model.

| Hyperparameter | Value |
| --- | --- |
| **MAE** | |
| ViT encoder size | depth: 4, heads: 4, embedding dim: 256 |
| ViT decoder size | depth: 3, heads: 4, embedding dim: 128 |
| Patch size | $8 \times 8$ |
| Mask ratio | 0.75 |
| Batch size | 1024 |
| Optimizer | Adam |
| Learning rate | 0.0003 |
| Pretrain step | 5000 |
| **RSSM** | |
| Deterministic state dim | 1024 |
| Stochastic state dim | 32 |
| Discrete latent dimensions | 32 |
| Batch size | 50 (Meta-world), 16 (DMC) |
| Sequence length | 50 |
| KL balance | 0.8 |
| Optimizer | Adam |
| Learning rate | 0.0003 |
| Gradient clip | 100 |

Table 4: Hyperparameters used in Actor Critic.

| Hyperparameter | Value |
| --- | --- |
| Replay buffer | 2,000,000 |
| Batch size | 50 |
| Trajectory length | 50 |
| Network size | [512, 512, 512, 512] |
| Optimizer | Adam |
| Learning rate | 0.0001 |
| Gradient clip | 100 |
| Entropy weight | 0.0001 |
| Discount | 0.99 |
| $\lambda$ return discount | 0.95 |
| Random steps | 5000 |
| Evaluate interval | 10,000 |
| Evaluate episodes | 10 (Meta-world), 5 (DMC) |

## H.1 ABLATION STUDIES

The importance of the exploration has been demonstrated in (Zhang et al., 2022). We perform ablation studies to demonstrate the effects of the major components, including representation dimension and window size, as illustrated below. Figure 5 presents an ablation study on representation dimension, where we compare $\mu$LV-Rep with latent representation dimensions 2048, 512, and 128. We also ablate the effect of window size $L$. In Figure 6, we compare $\mu$LV-Rep with window size $L = 1, 3, 5$. We also compare DrQ-v2 with $L = 1, 3, 5$ to show the effect of $L$ on other algorithms. The results show that for $L = 1$, both $\mu$LV-Rep and DrQ-v2 struggle with learning, which confirms the Non-Markovian property of the DMC control problems. We can also find that $L = 3$ is sufficient for learning in both test domains.

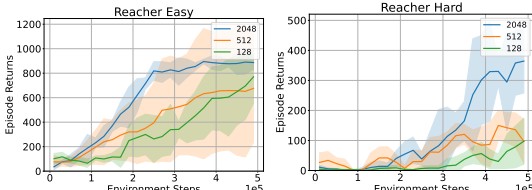

Figure 5: Ablation of feature dimension on visual control tasks from DeepMind Control Suites. Increasing the dimension of the feature gets better performance.

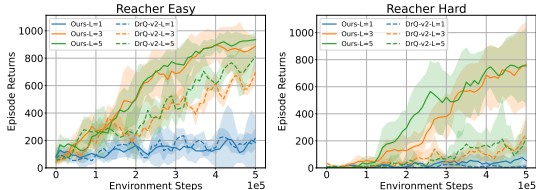

Figure 6: Ablation of window size $L$ on visual control tasks from DeepMind Control Suites. $L = 3$ is sufficient for learning in both test domains.