# OpenReview forum: "Provable Representation with Efficient Planning for Partially Observable Reinforcement Learning"
_ICLR.cc/2024/Conference — Submitted to ICLR 2024_

### Official Review · Reviewer_rUpE · 2023-10-26

**Soundness:** 3 good
**Presentation:** 1 poor
**Contribution:** 2 fair
**Rating:** 5
**Confidence:** 3

**Summary:**

This work aims to leverage the L-step decodable POMDPs to tackle the linear structure of POMDPs. Specifically, by conditioning on the recent L-step history (x_h), Q value does not need to rely on belief states or full history. Further, they show that Q value can be expressed in a linear form wrt P(z_h | x_h,a_h) where z_h is a latent variable, learned by some ELBO. The algorithm uses linear structure and learns the representation and Q value together with some sampling method. On both continuous and visual benchmarks, the proposed approach outperforms baselines in the terms of sample efficiency.

**Strengths:**

This work is original in POMDP literature with some theoretical guarantees (but I don’t have the expertise to check) and good quality. The technical writing is mostly clear, but some clarification is still needed.

The empirical results are persuasive that the proposed approach outperforms the other baselines in most domains in the chosen continuous and visual control benchmarks.

**Weaknesses:**

This work has a main issue in its story writing:

1. The title and abstract is quite vague and overly broad – basically it just said it is about a theoretical framework on RL or planning (I am confused which one) in POMDPs.

2. Moreover, the introduction on the theoretical framework is rather limited, unclear, unstructured, and seems overly strong. The 4 bullet points are most useful, but structured.

3. The claim “applied to a real-world problem” is especially strong, as the work obviously requires some assumptions on POMDPs, and no real world (like real robots) evaluation is performed.

4. The claim “state-of-the-art” is also too strong, as obviously Dreamerv2 and DrQ-v2, published in 2021, are no longer SOTA.

5. The claim also touches "offline POMDPs" but I did not see any results.

I don’t think this work has much technical significance since it is heavily relied on recent work (Ren et al, 2023a). Also, it would be better to point out how important linear structure is in solving POMDPs more explicitly.

**Questions:**

1. Is L-step decodability same as (or subsumed by) L-order MDPs?
2. Lack some definition on the latent variables z. From Eq 17, the objective of representation learning is exactly the same as belief-based approach. Let k=1, it is the standard ELBO of observation reconstruction/prediction, plus a KL divergence regularization between posterior and prior. In this sense, the optimal z seems to be belief state b. Is this correct? For k > 1, z might be different from b as it involves policy.
3. The paper talks about “low-rank POMDPs”, but no definition is provided. How is it connected to L-step decodability?
4. How partially observable is in the visual control benchmarks? As they were also tackled by Markovian methods.

---

> ### Author Response · Authors · 2023-11-19
> **Response to Reviewer rUpE**
>
> We provided some necessary introduction and common knowledge in RL and addressed the questions below.
>
> * RL vs. Planning
>
>   In RL,  the model learning refers to estimating the transition and reward,  and planning is referring to finding the optimal policy with the given or learned transition and reward (as explained in the Introduction section). We rigorously follow these definitions, where in our paper the algorithm **learns** a proper representation during the learning phase that can be used to perform planning (finding the optimal policy) easily.
>
> * Real-world problems
>
>   We tested our algorithm on MuJoCo environments with partial observations, and image-based RL problems, which are considered as real-world benchmarks in the RL community [1, 2]. We will rephrase this claim.
>   Again, we would emphasize the proposed  µLV-Rep is the first practical representation learning algorithm for POMDPs with tractable planning and exploration, and justified by rigorously  theoretical guarantees and strongly empirical performances, as far as we know.
>   It is unrealistic to ask one single conference paper to complete every aspect from theory, to simulation, to real robot deployment.
>
> * Offline POMDPs
>
>   The pessimism mechanism in offline setting is similar to the optimism in the online setting, and the major difference between the optimism in online setting and pessimism in offline setting is that, in the offline setting we **minus** the same bonus from the reward while in the online setting we **add** the bonus into the reward. We already added the corresponding discussion in Appendix D in our updated version.
>
> * Novelty and Significance comparing to Ren et al. (2023a)
>
>   We strongly disagree with this claim about the novelty and significance of our work.  Based on our personal communication with the authors of Ren et al (2023a), how to extend LV-Rep to POMDP is “extremely difficult, which cost them one year already”, as we also discussed in Section 4.
>   The extension is highly nontrivial and significant as we discussed in Section 3, especially considering the research on planning in POMDPs has stuck for decades. We emphasize the simplicity of the algorithm should be considered as an advantage, which enables the LV-Rep to more practical settings to handle partial observations, rather than a drawback.
>
> * L-step decodable POMDPs vs  L-order MDPs
>
>   We are not sure what is the concrete definition of L-order MDPs. We assume the reviewer is asking about the megastate MDPs defined in [3]. In fact, the connection between L-step decodable POMDPs and L-step megastate MDPs have been discussed extensively in [3].
>   One can convert L-step decodable POMDPs to a L-step megastate MDPs. This reduction is useful in justifying the difficulty of the problem by characterizing the lower bound dependency on the problem size. However, such a simple reduction will induce extra dependency in the megastate, as the successive megastates have overlap of the state trajectories. This extra dependency does not reduce the difficulty in planning, and it does not provide justification for low-rank linear representation in MDPs.
>   We understand the reviewer would like to use this reduction to justify the triviality of applying the existing representation learning for MDPs to POMDPs. However, as our personal communication and discussion with the authors of [1, 3, 4, 5], this argument does not hold.
>
> * Latent Variable z vs. Belief of state
>
>   No. Such understanding is incorrect.
>   Recall that by definition that belief is the posterior of the true state, while our latent variable z does not have to be the state but an arbitrary variable, as long as it satisfies the factorization in Eq. 16. Meanwhile, belief is a distribution, while z is a variable.
>   These properties of latent variable representation unleash us from identifiability restriction, which is one of our benefits to overcome the difficulty we discussed in Section 3 about recovering belief. We also discussed this in our main text in page 5 the remark about identifiability.
>
> * Low-rank POMDPs
>
>   The low-rank structure is indicating the latent variable factorization can be finite, or even if it is infinite, the eigendecay speed is fast, as we discussed in Appendix. The L-step decodability property ensures the rank is smaller than the exponential of the horizon, as discussed in [3].
>
> * Non-Markovian in Visual Control
>
>   In the image-based visual control benchmarks, although we can have the images, the control states, e.g., velocity, can not be inferred from one single image. There is no doubt that the Markovian methods can be applied for these tasks, but the performances will be significantly degenerated, due to the missing of important information.
>   To further demonstrate the non-markovian property, we tested the algorithms with markovian assumption (L=1) vs non-markovian (L>1) (Figure 6 of Appendix H.1). As we can see, the non-Markvoian algorithm performs significantly better.

---

> > ### Author Response · Authors · 2023-11-19
> > **References used in the Response**
> >
> > [1] Tongzheng Ren, Chenjun Xiao, Tianjun Zhang, Na Li, Zhaoran Wang, Sujay Sanghavi, Dale Schuurmans, and Bo Dai. Latent variable representation for reinforcement learning. In The Eleventh International Conference on Learning Representations, 2023b.
> >
> > [2] Tingwu Wang, Xuchan Bao, Ignasi Clavera, Jerrick Hoang, Yeming Wen, Eric Langlois, Shunshi Zhang, Guodong Zhang, Pieter Abbeel, and Jimmy Ba. Benchmarking model-based reinforcement learning. arXiv preprint arXiv:1907.02057, 2019.
> >
> > [3] Efroni, Yonathan, et al. Provable reinforcement learning with a short-term memory. In International Conference on Machine Learning, 2022.
> >
> > [4] Jiacheng Guo, Zihao Li, Huazheng Wang, Mengdi Wang, Zhuoran Yang, and Xuezhou Zhang. Provably efficient representation learning with tractable planning in low-rank pomdp. arXiv preprint arXiv:2306.12356, 2023
> >
> > [5] Qi Cai, Zhuoran Yang, and Zhaoran Wang. Sample-efficient reinforcement learning for pomdps with linear function approximations. arXiv preprint arXiv:2204.09787, 2022
> >
> > [6] Masatoshi Uehara, Ayush Sekhari, Jason D. Lee, Nathan Kallus, Wen Sun. Provably Efficient Reinforcement Learning in Partially Observable Dynamical Systems.   Neurips 2022 + arXiv preprint arXiv:2206.12020

---

> ### Comment · Reviewer_rUpE · 2023-11-20
>
> Firstly, I would like to address the tone of your response, which appears somewhat condescending. The phrase "We provided some necessary introduction and common knowledge in RL" suggests an assumption about the reviewer's level of expertise, akin to that of a student recently introduced to RL. Additionally, I noticed the absence of a customary 'thank you' in your response to my review, a courtesy extended to other reviewers. This inconsistency in approach is somewhat dismissive.
>
> However, as a professional reviewer, I still respond to your disconcerting rebuttal:
>
> > RL vs. Planning
>
> I definitely know what they are. So what is the planning part in your algorithm, like Dreamer's planning? The paper is too dense, as the other reviewer pointed out, and it's hard to grasp the core component of the algorithm.
>
> The problem on this point is writing. The title and abstract are too broad. The introduction focuses on the motivation and prior work too much.
>
> > By “efficient" we mean the statistical and computational complexity avoids an exponential dependence on history length, while the computational components of learning, planning and exploration are computationally feasible; while by “practical" we mean that every component of an algorithm can be easily implemented and applied to a real-world problem. In this paper, we provide a affirmative answer to these questions.
>
> Don't you feel these arguments are too strong? If so, RL is solved. I don't know anyone will call dm-control a real-world problem. To clarify, I don't ask you to work on real robots, but ask you to soften your overall writing. Unfortunately, you did not make any changes.
>
> > Offline POMDPs
>
> Ok, during rebuttal you have added it and the proof required the readers for some external references.
>
> > The claim “state-of-the-art”
>
> Unfortunately again, you did not respond to this.
>
> > Significance
>
> I'm not familiar with Ren et al.'s work. I also think the anecdotal evidence like personal communication is not a professional way to show some work is generally hard. Furthermore, the fact that the work is hard does not imply the work is significant.
>
> But at least, section 4.1 is very straightforward once you have L-step decodability assumption. Let the other reviewers and AC decide the main algorithm's significance.
>
> > L-order MDPs
>
> This concept stems from L-order HMM. You can view it as the recent L observations (and actions) in a POMDP compose of the state in the corresponding MDP.
>
> > Latent Variable z vs. Belief of state
>
> You did not directly answer my question. Can you compare Eq 17 with Dreamer's model learning objective? A belief state is a posterior, but it can be also viewed as a vector, expressed with the parameters of the posterior. In Dreamer or other variational belief approaches, they make it stochastic, i.e., we get a belief state z sampled from a distribution.
>
> > Low-rank POMDPs
>
> So L-step decodability belongs to low-rank POMDPs?
>
> > Non-Markovian in Visual Control
>
> Sounds good!

---

> ### Author Response · Authors · 2023-11-21
> **Further Response to Reviewer rUpE**
>
> Thanks for being professional. The following are our replies.
>
>
> * Significance
>   The claim that “Sec 4.1 is straightforward” after seeing the results lacks the basic respect not only for us, but also for the previous researchers, who are working on this problem.
>   The L-step decodability assumption has been proposed by Efroni et al, in 2020, and a recent followup paper trying to figure out computation tractable planning (Guo et al, 2023) is just published, while still leaving the planning problem partially open. If the planning problem is indeed “straightforward” as the reviewer claimed, the problem should be solved far earlier than our paper.
>   Again, we emphasize that simplicity does not mean straightforward. The simplicity of an algorithm should be considered as an advantage, rather than a drawback.
>
> * RL vs. Planning
>
>   * Re: Planning.
>
>     With the proposed representation, the Q-function can be represented linearly. And the planning is executing Bellman backup in the linear space as we described. While in Dreamer, it is not clear in which space the Bellman backup should be conducted to obtain a theoretical rigorous algorithm.
>
>    * Re: RL.
>
>      We never claim we solved RL problem, as we emphasize in the main text, the paper is trying to answer the question: “How can we design an efficient and practical RL algorithm for structured partial observations”, instead of solving RL, which we never claimed.
>
>      This question is right before the paragraph you cited. The paragraph you cited is written to explain the problem in detail. In fact, the proposed algorithm indeed avoids exponential dependence on history length, while with tractable planning and exploration. Meanwhile, if you read the paragraph you cited carefully, it claims that our algorithm **can be** applied to real-world problems. If you think there is any step of the proposed algorithm that cannot “be easily implemented and applied to a real-world problem“, please point it out and we are open to discuss.
>
> * L-order MDPs
>
>   In the RL community, this is known as L-step mega-state MDPs. We already explained the connection between L-decodable and L-step mega-state MDP in our last response, and more importantly, the difficulties in planning still exist even with the form of so-called “L-order MDPs”, due to the observation overlaps in mega-state.
>
> * Latent Variable z vs. Belief of state
>
>   We would like to clarify that belief refers to the posterior of **true state**, which requires extra identifiability assumption, while as we discussed in the remark in page 5.
>
>
> * Terminology confusion
>
>    * “Real-world problems”
>
>       We are sorry that the term “real-world” makes the reviewer uncomfortable. In fact, we already modified the claim in our experiment setting. While there are only three places in maintext, where the words show up:
>
>        * “In real-world reinforcement learning, state information is often only partially observable,” in abstract;
>        * “Such algorithms have been applied to many real-world applications with image- or text-based observations (Berner et al., 2019; Jiang et al., 2021), sometimes even surpassing human-level performance (Mnih et al., 2013; Kaufmann et al., 2023).” in page 1 introduction;
>        * “while by “practical" we mean that every component of an algorithm can be easily implemented and applied to a real-world problem.” in page 2.
>
>        If read carefully, you may see the first one is describing the problem setting, the second is describing dreamer and chatgpt, the third one is describing the target for which we are pursuing.
>
>     * "State-of-the-art”
>
>       We have softened our claim and removed “state-of-the-art”, which is used for our competitors, not our method.
>
> We sincerely believe the paper should be reviewed by evaluating the contribution and significance, instead of criticizing the usage of terminology.

---

> ### Comment · Reviewer_rUpE · 2023-11-21
>
> Thank you for your response. However, I must express that my primary concerns remain unaddressed. I find the tone of your response to be still condescending, as highlighted in the following points:
>
> >  “Sec 4.1 is straightforward”
>
> The phrase “Sec 4.1 is straightforward” was not meant to imply triviality. Simplicity in scientific approaches is often commendable and can be non-trivial. My intention was not to undermine the significance of your work, which I am uncertain about.
>
> Your criticism on me that "lacks the basic respect not only for us, but also for the previous researchers, who are working on this problem." is unfounded and quite troubling. Such an accusation is not conducive to a constructive scientific discourse.
>
> > We never claim we solved RL problem, as we emphasize in the main text, the paper is trying to answer the question.
>
> When I referred to your strong claims, it was my interpretation of the tone and implications in your paper, not an accusation that you claimed to have "solved the RL problem".
>
> > cannot “be easily implemented and applied to a real-world problem"
>
> To be specific, you can check the assumptions made in your approach. For example, the definitions 1,2 are on the structures of POMDPs. How does your algorithm scale to very large L, which is common in real-world problem?
>
> > L-order MDPs
>
> Thanks and I got it now. However, your previous claim hat
>
> > We understand the reviewer would like to use this reduction to justify the triviality of applying the existing representation learning for MDPs to POMDPs. However, as our personal communication and discussion with the authors of [1, 3, 4, 5], this argument does not hold.
>
> Your assumption that my critique is intended to trivialize is again groundless. My goal was to seek clarity.
>
> > Belief of state
>
> To clarify, I was asking about approximate belief state, like what these variational approaches learn.
>
> > "State-of-the-art”
>
> I did not see that you have updated your PDF.
>
> > We sincerely believe the paper should be reviewed by evaluating the contribution and significance, instead of criticizing the usage of terminology.
>
> It is essential to support claims of being SOTA with solid empirical evidence. The usage of such terminology is not just a semantic issue; it reflects the scope and impact of your claims.
>
> Finally, my objective is not to criticize but to ensure the clarity, accuracy, and significance of your contributions to the field.

---

> > ### Author Response · Authors · 2023-11-22
> > **Responese to Reviewer rUpE**
> >
> > Thanks for clarifying the questions and claims in the reviews.
> >
> >
> > >“Sec 4.1 is straightforward” in the reviewer’s reply and “I don’t think this work has much technical significance” in original review
> >
> > We believe “straightforward”  and no “technical significance” might not be appropriate to be used here, which may incur unnecessary misunderstanding. We are glad that we are finally on the same page that solving hard problems in a “simple” way is elegant, as we emphasize iteratively.
> >
> >
> > >“scale to very large L”
> >
> > We first clarify that if the problem is with large L, then, it is the inherent hardness of the problem, rather than the algorithm complexity. As in computational complexity theory, we should not blame the computational/memory complexity of an algorithm that solves an NP-hard problem, due to the essential difficulty of the NP-hardness. Similarly here, we should not blame the complexity of the algorithm aiming for planning over  large L POMDPs, because that is the essential difficulty of the problems.
> >
> > Secondly, even for large L, we have several practical alternatives:
> > * i), use some appropriate length for approximation;
> > * ii), increase the RNN, which can encode the whole history theoretically.
> >
> > In fact, as we demonstrated in the newly added ablation study, for image-based control, 3-5 frames are enough. Even for text-based agents, e.g., ChatGPT, we can still handle the problems with a reasonable L.
> >
> >
> > >“Belief of state”
> >
> > We have discussed this in the main text and our previous response, and we will explain here again.
> >
> > L-step decodability only ensures that the beliefs only depend on L-step observations rather than the whole history, but does not guarantee that the beliefs are “identifiable”. The “non-identifiability” means there are potentially many distributions of latent variables that achieves the maximum of the optimization, but only one solution is about the distribution of ground-truth state, which is known as beliefs. Without any additional assumption, even with L-step decodability, it is not clear which solution is the belief. Therefore, it s not clear which one to approximate.
> >
> > Moreover, as we discussed in Sec 3, even with some approximation of the beliefs, there will be extra difficulties in representing $Q$, therefore, difficulties in planning and exploration.
> >
> > In fact, one of our contributions is that we reveal the fact that there is no need to obtain or approximate beliefs, any valid factorization is sufficient for representation $Q$-function.
> >
> >
> > >“State-of-the art”
> >
> >
> > Thanks for your suggestions. Please take a look at the revised draft yesterday at 20 Nov 2023, 20:43 Eastern Standard Time, where we already updated following your suggestion.

---

> ### Comment · Reviewer_rUpE · 2023-11-22
> **My main concerns are not resolved and I maintain my score**
>
> Thanks for your replies. As the discussion is close to end, I do not expect further responses to my following comment and summary. I believe with additional efforts, your work has the potential for publication in a top-tier conference.
>
> > We believe “straightforward” and no “technical significance” might not be appropriate to be used here.
>
> I'd like to clarify my previous statement: I mentioned the work has "not much significance", not "no significance". I trust this misquotation was accidental.
>
> Regarding the use of "straightforward", I maintain its appropriateness. The key observations, as the paper stated, are straightforward in replacing belief state with history, and history with $L$-step memory, based on your assumptions.
>
> > the claim of real-world problem, large $L$
>
> Concerning your assumption on $L$-step and its real-world applicability, testing with $L$ values between 2 to 5 is insufficient to claim easy implementation in real-world problems without empirical evidence. The validity of the linear structure assumption in real-world problems also remains unclear.
>
> I recommend softening the claim about real-world applicability to "has the potential to be applied", unless you provide empirical support.
>
> > “Belief of state”
>
> Let me be straightforward. Here I rewrite Eq. 17 with $l=1$:
> $$
> \max_q E_{q(z_h \mid x_h,a_h,o_{h+1})}[\log P(o_{h+1} \mid z_h)] - D_{KL}(q(z_h \mid x_h,a_h,o_{h+1}) \mid\mid p(z_h\mid x_h))
> $$
> The first term is to reconstruct observations, and the second term is KL divergence between the posterior of $z$ and prior of $z$.  The DreamerV2 model learning objective is very similar to this, except that they have a reward prediction term (which I am unsure about your method) and the prior also conditions on $a_h$ (which I believe should be added).
>
> > state-of-the-art
>
> I found you changed it in the main PDF, but not supplementary material. Please also update the supp accordingly.
>
> Finally, I want to conclude my review. I maintain my rating due to the seemly limited significance of the work, dense writing style, unclear contributions and connections with prior work, some overreaching claims, and the tone and unprofessionalism of responses to my feedback (e.g., several anecdotal evidences and assumed negative intent on the reviewer) which I flagged for future ethics review.

---

> > ### Author Response · Authors · 2023-11-23
> >
> > Well, this close to the end of the discussion. It is really nice to discuss with you. Thanks for your valuable comments.
> >
> > > "Straightfroward"
> >
> > First, it is not appropriate to claim something is straightforward after seeing the results.
> >
> > The replacing with belief with history is straightforward, as you mentioned. But identifying the "linearity" in Q w.r.t. representation is high non-trivial, in our derivation from Eq. 11 to Eq. 15, which is emphasized in main text as "our most important observation", but totally ignored by you (I trust this neglection was accidental, rather than intentional).
> >
> > > "Large L "
> >
> > We have replied this in our previous response.
> >
> > This is the inherent hardness of the problem, rather than the algorithm complexity. We also provided tractable solution for approximation.
> >
> > Meanwhile, we already demonstrated the potential in lots of envs in several mainstream benchmarks, including MuJoCo, DMC, and Meta-world.
> >
> > Your claim **2-5 is insufficient in practice** is slightly contrastive to your previous claim that image-based control can be solved by **markoviann** setting, which is equivalent to L=1.
> >
> > > "Belief of state"
> >
> > We have introduced several times about the differences between belief and latent variable. Let's put the difference between beliefs and latent variable aside.
> >
> > The major difference in Dream and the proposed method is how we use the obtained latent variable representation. Due to the linearity we revealed, the Q function can be represented in the latent variable distribution space, as we explained in Eq. 18. This representation enable the efficient planning and exploration, which has not been discussed in Dreamer.
> >
> > Finally, we believe everyone working in RL is aware of the gap between theoretical work and empirical work. We hope the community could appreciate the work that is at least trying to close such gaps.
> >
> > Happy Thanksgiving.

---

### Official Review · Reviewer_wQNF · 2023-10-31

**Soundness:** 2 fair
**Presentation:** 1 poor
**Contribution:** 3 good
**Rating:** 5
**Confidence:** 3

**Summary:**

This paper contributes a new algorithm for RL in structured POMDPs.
It proposes to use a latent variable model to learn a linear representation of the value function in L-step decodable POMDPs.
The proposed approach shows good performance in a large set of tasks when compared with baselines.

**Strengths:**

- The work is highly relevant for the RL community, as it explicitly tackles problems with partial observability, a fundamental challenge for applying RL in real-world tasks.

- The method proposed is relatively novel, as it combines efficient linear representations with L-step decodable POMDPs.

- The empirical evaluation considers many tasks and shows the proposed method has strong performance compared with multiple baselines.

**Weaknesses:**

- The presentation could be improved. Some technical details are inconsistent or lack an appropriate definition (see detailed comments below). This makes important parts of the paper challenging to comprehend, such as the discussion of Eq 9.

- The paper is also very dense, which makes some parts too condensed. For instance, the theoretical analysis only states the assumptions and an informal version of the sample complexity of the algorithm without including an analysis of this result.

- The empirical evaluation is limited to a comparison with other algorithms. It would be interesting to provide an ablation study to show how the different components of the algorithm contribute to its performance. For example, how the algorithm performs without optimistic exploration.
Furthermore, it would be interesting to make a hyper-parameter sensitivity analysis, for example, evaluating how the algorithm performs with different values of L.


[Detailed comments]
- wrong typesetting of the observation function in the first paragraph of the preliminaries
- in the preliminaries, should the agent receive a reward r(s_h, a_h)?
- In the belief definition, it is unclear what is P(s1\mid o1). It is also unclear what is \tau.
- Wrong index in the actions of Eq 2
- Unclear what is \theta on Eq 5?
- \mu is used for initial state distribution and as a feature map
- are Eq 6 and Eq 9 missing some <> delimiters?
- after Eq 7: an practical -> a practical
- Eq 8: R(s,a) -> r(s,a)
- Eq 2 and 3 are defined for problems with a finite horizon, then Eq 6 and 8 use discount factor $\gamma$.
- Eq 12 uses the latent variable z without a proper introduction

**Questions:**

1. After Eq 16, could you provide some intuition about what is the parameter l?

2. Could you provide a formal definition of the policy \mu_pi?

3. The experimental evaluation mentions that the algorithms were tested after running 200K environment steps. Could you comment on this choice of training time? In particular, is this training budget sufficient for the convergence of all algorithms?

---

> ### Author Response · Authors · 2023-11-19
> **Response to Reviewer wQNF**
>
> We thank the reviewer for the suggestion on the presentation and other feedback. We address the concerns in the following:
>
> * Presentation
>
>   Despite the complicatedness of the related techniques and the subtleness in planning for POMDPs, which makes the paper technically dense, we try our best to make the paper self-contained, and design a simple algorithm.
>   We have revised our manuscript accordingly and try our best to make the paper more convenient to be understood. We would like to clarify some key typos in the previous manuscript
>
> * * Reward $r(s_h, a_h)$
>
>   In fact the agent can only observe a reward $r(o_h, a_h)$, which can be defined as $$r(o_h, a_h) = \int r(s_h, a_h) P(o_h|s_h) ds_h$$ and can be viewed as a sample from $r(s_h, a_h)$, which is commonly assumed in the existing POMDP literature [1, 2]
>
> * * $P(s_1|o_1)$ and $\tau$
>
>     $P(s_1|o_1)$ can be defined as the posterior distribution induced by the initial distribution and emission. $\tau$ is the history defined in the second paragraph of the preliminaries.
>
> * * $\theta$ in Equation 5
>
>     $\theta$ is just a vector that can induce the reward.
>
> * Parameter l in Eq. 16
>
>   For any $l \in N_+$, $p({o_{h+1:h+l}}|x_h, a_h) \int_{\mathcal{Z}} p(z_h|x_h, a_h)  p^\pi ({o_{h+1:h+l}}|z_h) d z_h$ Is factorizable and share the same $p(z_h|x_h, a_h)$, but with different  $p^\pi ({o_{h+1:h+l}}|z_h) $.
>   As we discussed that  $p(z_h|x_h, a_h)$ forms the function space for Q-function, and thus, is the latent variable representation we are seeking, we can perform MLE on arbitrary l-step future observations conditional distribution.
>
> * Definition for the moment matching policy $\nu_\pi$
>
>   The formal definition of νπ is complicated and we provide a formal definition at the beginning of the proof for Lemma 8. We also add more concrete discussion in Appendix C about the moment matching policy.
>
> * Ablation Study
>
>   We have added more ablation studies (Appendix H.1) to demonstrate the effects of the major components, including representation dimension and window size.
>
> * Experimental setting
>
>   We set up the experiment setting exactly following the benchmark [1], which has been widely adopted in [2,3,4] for fairness, especially for the comparison to LV-Rep with full observation in Table 1 (denoted as Best-FO), to demonstrate the performance gap in MDP and POMDP.
>
> [1] Tingwu Wang, Xuchan Bao, Ignasi Clavera, Jerrick Hoang, Yeming Wen, Eric Langlois, Shunshi Zhang, Guodong Zhang, Pieter Abbeel, and Jimmy Ba. Benchmarking model-based reinforcement learning. arXiv preprint arXiv:1907.02057, 2019.
>
> [2] Tianjun Zhang, Tongzheng Ren, Mengjiao Yang, Joseph Gonzalez, Dale Schuurmans, and Bo Dai. Making linear mdps practical via contrastive representation learning. In ICML, 2022
>
> [3] Tongzheng Ren, Chenjun Xiao, Tianjun Zhang, Na Li, Zhaoran Wang, Sujay Sanghavi, Dale Schuurmans, and Bo Dai. Latent variable representation for reinforcement learning. In The Eleventh International Conference on Learning Representations, 2023b.
>
> [4] Tongzheng Ren, Tianjun Zhang, Lisa Lee, Joseph E Gonzalez, Dale Schuurmans, and Bo Dai. Spectral decomposition representation for reinforcement learning. ICLR, 2023c.

---

### Official Review · Reviewer_cvkF · 2023-11-01

**Soundness:** 2 fair
**Presentation:** 2 fair
**Contribution:** 2 fair
**Rating:** 6
**Confidence:** 3

**Summary:**

While Partially Observable Markov Decision Processes (POMDPs) were introduced to address partial information in RL algorithms where full
observability is unavailable, such formulation brings computational challenges in learning, exploration, and planning, due to the non-Markovian dependence between observations. This paper aims to address the computational and statistical challenges in Partially Observable Markov Decision Processes (POMDPs). In particular, the authors develop a representation-based perspective that leads to a coherent framework and tractable algorithm.

**Strengths:**

This paper studies the problem of designing an efficient and practical RL algorithm for structured partial observations in RL frameworks. Authors introduce a structured POMDP with a low-rank property that allows for a linear representation, called Multi-step Latent Variable Representation (µLV-Rep), which is a counterpart of linear MDP in the POMDP context. As such, this representation overcomes computational barriers and enables a tractable representation of the value function. The extension of linear MDP to POMDP can be beneficial.

The paper also proposes a planning algorithm that can implement both the principles of optimism and pessimism in the face of uncertainty for online and offline POMDPs.

Theoretical analysis in sample complexity and PAC guarantee are provided to justify the performance guarantee.

Empirical comparisons are performed to demonstrate the performance on a set of benchmark environments compared to existing SOTA RL algorithms for POMDPs.

**Weaknesses:**

1. The theoretical analysis relies on quite a few assumptions (including (Finite Candidate Class with Realizability, Normalization Conditions, Regularity Conditions and Eigendecay Conditions), which may not always fulfilled in reality. Can authors comment on the performance of the algorithms when these assumptions break, e.g., how worse the performance is going to be, and which of the assumptions are essential to retain the performance?

2. The theoretical analysis is mainly based on Ren et al., 2023a. It is unclear what are the technical novelties in the analysis compared to Ren et al., 2023a. Authors are expected to explain the difference and highlight the key insights in the proofs.  In particular, the proof of Theorem 12 is unclear by just claiming "This is a direct extension of the proof of Theorem 9 in Ren et al. (2023a)". The technical contribution in theory remains questionable.

3. Algorithmically, the proposed main algorithms borrow lots of the elements from Ren et al., 2023a, the novelty appears to be limited.

4. In section 4, it is unclear how the planning algorithm implements pessimism for offline RL.

5. Can authors comment on the tightness of the sample complexity bounds in Lemma 11?

**Questions:**

See above. In addition, there are some minor grammatical errors in the draft. It is suggested that authors carefully proofread the draft for improvement.

---

> ### Author Response · Authors · 2023-11-19
> **Response to Reviewer cvkF**
>
> We thank the reviewer for the feedback. Please see our response below:
>
> * Assumptions
>
> We first emphasize that the assumptions are merely for the proofs, which are commonly used in the literature (e.g. [1, 2, 3, 4]) and are thought to be mild. While the proposed µLV-Rep is still  widely applicable even without the assumptions in practice. In fact, in our empirical experiments, the assumptions are not guaranteed to be held. But the proposed algorithm still outperforms or is comparable to the existing strong competitors, including DreamerV2, which demonstrates the robustness of the proposed algorithm in practice.
>
> * Novelty and Significance comparing to Ren et al. (2023a)
>
> As we mentioned, our core contribution is not theoretical analysis. Instead, our contribution lies on extending LV-Rep for POMDP that can perform provable representation learning and planning with function approximations (as illustrated in Section 4), which is highly nontrivial and significant as we discussed in Section 3, especially considering the research on planning in POMDPs has stuck for decades.
> We emphasize the simplicity of the algorithm should be considered as an advantage, which enables the LV-Rep to more practical settings to handle partial observations, rather than a drawback.
>
> * Pessimism in Offline
>
> The pessimism mechanism in offline setting is similar to the optimism in the online setting, and the major difference between the optimism in online setting and pessimism in offline setting is that, in the offline setting we **minus** the same bonus from the reward while in the online setting we **add** the bonus into the reward. We already added the corresponding discussion in Appendix D our updated version.
>
> * Tightness of Bounds
>
> First, we emphasize that our sample complexity bound is for POMDPs, which matches the lower bound dependency on the horizon of actions [1]. Regarding the tightness of the dependency on the other quantities in the sample complexity, it is still an open problem even for  the existing representation learning methods including [2, 3, 4].
> The proposed  µLV-Rep is the first practical representation learning algorithm for POMDPs with tractable planning and exploration, with strong theoretical guarantees, as far as we know. How to provide better representation learning algorithms with better sample complexities is still an open problem and beyond the scope of this manuscript.
>
> [1] Efroni, Yonathan, et al. Provable reinforcement learning with a short-term memory. In International Conference on Machine Learning, 2022.
>
> [2] Agarwal, Alekh, et al. ”Flambe: Structural complexity and representation learning of low rank mdps.” Advances in neural information processing systems 33 (2020): 20095-20107.
>
> [3] Uehara, Masatoshi, et al. ”Representation Learning for Online and Offline RL in Low-rank MDPs.” International Conference on Learning Representations. 2022.
>
> [4] Ren, Tongzheng, et al. ”Latent Variable Representation for Reinforcement Learning.” The Eleventh International Conference on Learning Representations. 2023

---

### Author Response · Authors · 2023-11-19
**General Response**

We would like to thank reviewers for their work during the review process. However, we believe there is a common misunderstanding across the reviewers on the difficulty of the target problem, i.e., provable planning for POMDPs, and thus, the significance of our paper.

We would like to emphasize that our focus is identifying a practical subclass of POMDPs that have proper linear representations, and designing computational tractable planning, learning and exploration algorithms. Such a problem can be notoriously difficult as we discussed in Section 3, which has had no concrete progress since (Jolle et.al., 2009) with function approximator for more than a decade. The recent work (Golowich et. al., 2022a, b) investigated this problem with similar assumptions (L-observable POMDPs), but still under tabular cases, and thus, not applicable for practical problems.

We successfully solve the problem and push the understanding and usage of POMDPs. Our contributions lies in three-folds:
* We first successfully identify a subclass of POMDPs that can perform efficient planning with a proper linear representation.
* We then extend the existing LV-Rep framework to learn such representation and form a practical algorithm with rigorous guarantees. Such an extension is highly **non-trivial**, as we discussed in Section 3.
* More importantly, we empirically justify the performances of the proposed algorithms, with the comparison with the current existing algorithms.

We emphasize the simplicity of the algorithm should be considered as an advantage, which enables the LV-Rep to more practical settings to handle partial observations, rather than a drawback.

We hope such a clarification can help to make our core contribution and significance clear, especially under the understanding of the difficulty in tractable planning and learning for POMDPs.

We modify the draft (with blue color texts) to address the concerns accordingly.

---

### Meta-Review · Area_Chair_VraX · 2023-12-06

**Metareview:**

This paper follows up on the effort of the recent work [1] in designing statistical and computational tractable algorithms for POMDPs. Many of the computational techniques are adopted from [2] but still a significant amount of innovations and key insights went into making a computationally tractable algorithm possible, solving some of the remaining issues in [1].

Judging as someone who is very familiar with the line of theory work on POMDPs, I believe the technical contributions in this paper is solid. However, the presentation of the paper can be significantly improved. The current presentation (especially of section 4.1 where the authors motivate and describes their main technical innovations) is only understandable to a handful of readers who are authors to or familiar with the recent line of theoretical RL work on POMDPs and related work such as [1] and [2]. For a reader experienced with RL theory but not POMDPs or for a reader from the empirical side, the main text of this paper is too dense and lack of context. In fact, most of the reviewers had trouble understanding and appreciating the key technical contribution of the paper.

Therefore, I recommend the authors to make efforts in improving the presentation of the paper.


[1] Jiacheng Guo, Zihao Li, Huazheng Wang, Mengdi Wang, Zhuoran Yang, and Xuezhou Zhang. Provably efficient representation learning with tractable planning in low-rank pomdp. arXiv preprint arXiv:2306.12356, 2023.

[2] Tongzheng Ren, Chenjun Xiao, Tianjun Zhang, Na Li, Zhaoran Wang, Sujay Sanghavi, Dale Schuurmans, and Bo Dai. Latent variable representation for reinforcement learning. In The Eleventh International Conference on Learning Representations, 2023b.

Note from SAC: I want to give some more concrete suggestions regarding improving the paper. Take Section 4.1 as an example: the section provides a sequence of "derivation" steps, but the purpose of these derivations was very unclear from the text (other than the vague claim of computational efficiency). Ideally it should be broken down into a number of small lemmas and propositions that are useful for later proofs. Relatedly, it will be very helpful to explicit state at the beginning of the section, in very clear and precise mathematical language, what is the end goal of all these derivations? Then at the end of the derivation, the paper should confirm, with lemmas/propositions/theorems, that such a goal is achieved. If computational efficiency is the goal, one can start by reviewing why the problem, in its original form, is not computationally efficient, and what are the major obstacles. Then, during derivations, it should be highlighted if a transformation step alleviates certain obstacles. At the end, if full efficiency is achieved, a theorem/proposition should be given, potentially with a computational complexity analysis (in what sense is the algorithm efficient? do you need oracles? what kind? etc.).

**Justification For Why Not Higher Score:**

NA

**Justification For Why Not Lower Score:**

NA

---

### Decision · Program_Chairs · 2024-01-16

Reject